

# On the Resonance Hypothesis of Tsunami and Storm Surge Runup

Nazmi Postacioglu [1], M. Sinan Özeren [2], and Umut Canlı [1]

[1]Istanbul Technical University, Department of Physics, Maslak 34469 İstanbul, Turkey
[2]Istanbul Technical University, Eurasia Institute of Earth Sciences, Maslak 34469 İstanbul, Turkey

*Correspondence to:* M. Sinan Özeren (ozerens@itu.edu.tr)

**Abstract.** Resonance has recently been proposed as the fundamental underlying mechanism that shapes the amplification in coastal runup for both Tsunamis and storm surges. It is without doubt that the resonance plays a rôle in runup phenomena of various kinds, however we think that the extent at which it plays its role has not been completely understood. For incident waves, the best approach to investigate the rôle played by the resonance would be to calculate the normal modes by taking

radiation damping into account and then test how those modes are excited by the incident waves. There are a small number of previous works that attempt to calculate the resonant frequencies but they do not relate the amplitudes of the normal modes to those of the incident wave. This is because, by not including radiation damping, they automatically induce a resonance that leads to infinite amplitudes, thus preventing them from predicting the exact contribution of the resonance to coastal runup. In this study we consider two different coastal geometries: an infinitely wide beach with a constant slope connecting to a

flat-bottomed deep ocean and a bay with sloping bottom, again, connected to a deep ocean. For the fully 1-D problem we find significant resonance if the bathymetric discontinuity is large. For the 2-D ocean case the analysis shows that the wave confinement is very effective when the bay is narrow. The bay aspect-ratio is the determining factor for the radiation damping.

## 1 Introduction

During the last decades, several analytical and numerical studies of coastal runup were published. The majority of the analytical

studies were conducted on 1-D profiles (see Synolakis (1987), Carrrier et al. (2003), Kânoğlu (2005), Kânoğlu and Synokalis (2005), Özeren and Postacioglu (2012), and Stefanakis et al. (2015)) most of which made use of Carrier-Greenspan transformations (Carrrier et al. (2003)). Some of these works, for instance Stefanakis et al. (2015), Stefanakis and Dias (2011) and Ezersky et al. (2013b), identified the resonance as the fundamental factor for the runup amplification. The bulk of the present study will be dedicated to determine those physical settings in which this might be the case.

In the past, several researchers looked at resonance aspect of the coastal runup. Among those, the ones that are the most relevant to the discussion in the present study are Stefanakis et al. (2015), Stefanakis and Dias (2011), Carrier and Noiseux (1983), Ezersky et al. (2013a), Fuentes et al. (2015), Volker et al. (2010) and Yamazaki and Cheung (2011). The last two of these studies report coastal resonance mechanisms leading to amplified runups during the 2009 Samoa and 2010 Chile Tsunamis respectively. The work by Carrier and Noiseux (1983) does not explicitly mention *resonance* but their formulation

clearly shows a constructive interference by multiple reflections of an obliquely incident Tsunami wave for a particular set




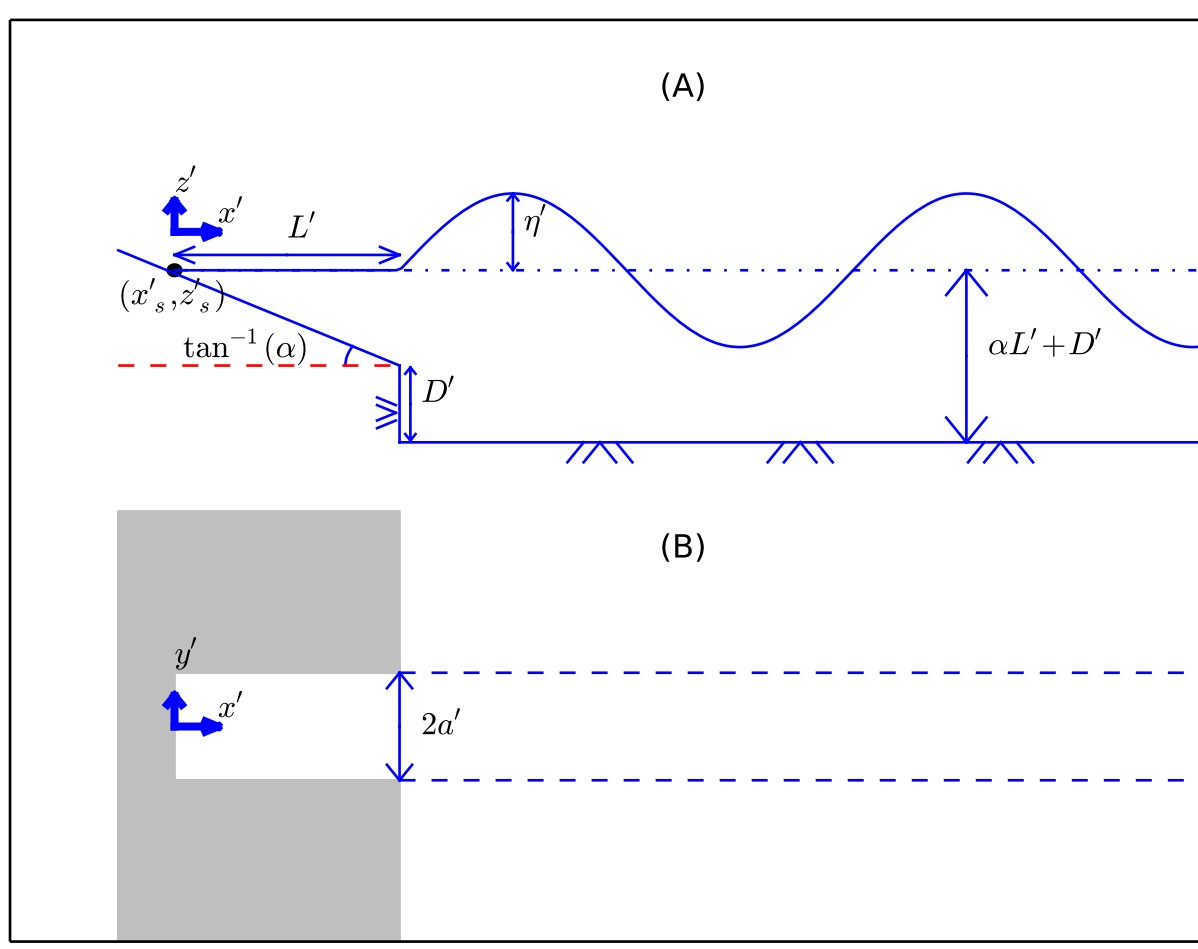

**Figure 1.** (A) is the side view of the the incident wave at instant $t' = 0$. The black dot is the shoreline and $(x'_s, z'_s)$ are its coordinates with $z'_s$ being equal to runup $r'$. (B) is the geometry of the channel seen from above. In *MODEL-1* the sloping channel is connected to a deeper channel of the same width (see broken lines in (B)). In *MODEL-2* the channel opens to a semi-infinite ocean. The width of the channel is $2a'$. Non- dimensional quantities are defined as $x = x'/L'$, $y = y'/L'$, $z = z'/(\alpha L')$, $\eta = \eta'/(\alpha L')$, $D = D'/(\alpha L)$ and $a = a'/L'$.



of incidence angles. Some of the studies looked at the resonant mechanism in experimental settings (Ezersky et al. (2013a), Abcah et al. (2016)).

In our modelling, we will be considering monochromatic incident waves, simply because this makes it easier to shed light on the resonance. However, the mathematical algorithm we will develop is not limited to monochromatic waves but is capable

of calculating runup for any kind of offshore source including the earthquakes, submarine landslides or atmospheric pressure perturbations. The only limitation is that the amplitude of the incident wave should be small enough to allow linearization at the toe of the coastal slope, a very common assumption in non-perturbative analytical studies of coastal wave phenomena.

This article deals with the *transient* runup response of a sloping channel (or bay) to an incident wave. Here the term *transient* is used in the sense that the monochromatic wave starts to be generated at a given instant, thereby presenting an initial value

problem for an infinite domain. Such an initial value problem can be difficult and expensive to handle by purely numerical approaches. The reason for the difficulty is that, on the offshore boundary it is difficult to make a distinction between the incident and reflected waves. One way of surmounting this difficulty numerically is to take the computational domain so large that by the time the reflected waves arrive at the offshore boundary, the steady regime would have set in the coastal region. For instance, Stefanakis and Dias (2011) did one-dimensional numerical simulations to understand the runup amplification by

nonleading long waves. They cast the problem as a boundary value problem and imposed an offshore boundary condition, at distance $L'$ from the undisturbed shoreline, for the wave height as $\eta' = \pm \eta'_0 \sin(\omega' t')$ where $t'$ is time. However, this model tends to overestimate the runup if the open boundary lies in one of the nodes of the standing wave that will eventually set in. The problem with this approach is that the incident wave may interact, constructively or destructively, with the reflected waves unless one takes the distance $L'$ large enough and this is not the case in Stefanakis and Dias (2011). When the incident and

reflected wave distinction *is* made (see, for example, Antuono and Brocchini (2010)), the runup amplification factor (defined as $r'/(2\eta_0'^I)$ where $\eta_0'^I$ is the incident wave amplitude and $r'$ is the runup) remains finite as long as the frequency of the incident wave is real. This has independently been shown in Antuono and Brocchini (2010).

Our purpose in the present work is to determine the way in which the free modes near the coast are excited by the incident waves by taking the radiation damping into account. We will examine resonance in two different geometric settings: first a 1D

slope which connects to a 1D channel with a flat bottom and then, again a 1D slope that connects to a semi-infinite 2D ocean with a flat bathymetry (see Figure 1). When the wavelength of the incident wave is much shorter than the width of the sloping channel the completely 1D model is a good approximation to the natural case and we can neglect the geometric spreading of the waves at the toe of the sloping bay (this geometry will be referred to as *MODEL-1* during the rest of the article). If this is not the case, then a 2D model (*MODEL-2*) near the mouth of the bay becomes necessary (see Figure 1). The Coriolis acceleration

is neglected in both cases as we limit ourselves to small scales.

*MODEL-1* is actually solvable, through fast Fourier transforms (see Ezersky et al. (2013b)), without necessarily resorting to the coastal free modes. However generalizing this solution approach to 2D in the deep ocean part is computationally very expensive, requiring a solution of an integral equation for each frequency component to calculate the transient response. Hence, the real importance of the technique developed in this article becomes more apparent in *MODEL-2* which can be of great





engineering importance in modelling coastal amplifications of Tsunamis and storm surges in places like the Tokyo Bay which has a sloping bathymetry (Kataoka et al. (2013)).

## 2 Basic equations

*MODEL-1* consists of a channel of constant slope $\alpha$, length $L'$ that connects to a another channel of uniform depth $\alpha L' + D'$

(see Figure 1). Parameter $D'$ is the discontinuity in depth at toe of the slope. The maximum depth of the sloping channel is $\alpha L'$
.

The governing equations we shall use over the sloping part of the geometry are non-linear shallow water equations

$$\partial_{t'} u' + u' \partial_{x'} u' + g \partial_{x'} \eta' = 0 \tag{1}$$

$$\eta'_{t'} + \partial_{x'} \left( (\alpha x' + \eta') u' \right) = 0 \tag{2}$$

where $u'$ is depth-averaged velocity in offshore-pointing $x$ direction, $t'$ is time, $g'$ is the acceleration due to gravity and $\eta'$ is the wave height. For the flat part of the domain, for *MODEL-1*, the linearized versions of the same equations,

$$\partial_{t'} u' + g \partial_{x'} \eta' = 0 \tag{3}$$

$$\eta'_{t'} + \partial_{x'} \left( (\alpha L' + D') u' \right) = 0. \tag{4}$$

are used. Note that, when we shall eventually start to discuss *MODEL-2*, we will generalize equations (3) and (4) into two

dimensions. Let us now define non-dimensional quantities as

$$
\begin{aligned}
x &= x'/L', \quad \eta = \eta'/(\alpha L') \\
t &= t'\sqrt{\alpha g'/L'}, \quad u = u'/\sqrt{g'\alpha L'}, \quad D = D'/(\alpha L').
\end{aligned} \tag{5}
$$

A hodograph transformation introduced by Carrrier et al. (2003), also called Carrier-Greenspan transformation, uses "distorted time" $\lambda$ and a potential $\varphi$ defined as

$$\lambda = t - u \tag{6}$$

with

$$u = -\partial_\sigma \varphi/(2\sigma) \,, \; \eta = \partial_\lambda \varphi - (\partial_\sigma \varphi)^2/(8\sigma^2) \tag{7}$$




where

$$\sigma = \sqrt{x + \eta}. \tag{8}$$

The non-linear shallow water equations accordingly become (Carrrier et al. (2003))

$$\partial_{\lambda\lambda}^2 \varphi - \frac{1}{4\sigma} \partial_\sigma \varphi - \frac{\partial_{\sigma\sigma}^2 \varphi}{4} = 0 \, . \tag{9}$$

To treat the incident wave problem, the initial conditions everywhere are

$$\eta(t=0,x) = \eta^I_{\text{initial}} \tag{10}$$

and

$$u(t=0,x) = \frac{-\eta^I_{\text{initial}}}{\sqrt{D+1}} \tag{11}$$

for the wave height $\eta$ and the fluid velocity $u$ respectively. We assume that both quantities are initially zero over the slope
$(0 < x < 1)$. The minus sign in (11) is due to the fact that the progressive wave advances in the negative-$x$ direction towards
the coast on the left.

## 3   Green's function and the free mode expansion

For the flat part of the domain $(x > 1)$ we propose the following solution

$$\eta(t,x>1) = \int_{-\infty}^{\infty} \left( \tilde{\eta}^I(\omega) \exp\left( i\omega \left( t + \frac{x-1}{\sqrt{D+1}} \right) \right) + R(\omega) \exp\left( i\omega \left( t - \frac{x-1}{\sqrt{D+1}} \right) \right) \right) d\omega \tag{12}$$

where $\tilde{\eta}^I(\omega)$ and $R(\omega)$ are the temporal Fourier transforms of the incident and the reflected waves (to be calculated) respectively. Here $\tilde{\eta}^I_\omega$ is defined as

$$\tilde{\eta}^I(\omega) = \frac{1}{2\pi} \int_{-\infty}^{\infty} \eta^I(t,x=1) \exp(-i\omega t) dt. \tag{13}$$

The integrand in the right-hand side of (13), for all times, can be inferred from the initial condition as $\eta^I(t,x=1) = \eta^I(0,1+t\sqrt{D+1})$. Note that the choice of the point $x=1$ is completely arbitrary. Let us now propose a solution over the slope as

$$\varphi = \int_{-\infty}^{\infty} A_0(\omega) J_0(2\omega\sigma) \exp(i\omega\lambda) d\omega \tag{14}$$

The corresponding wave height near $x=1$ (toe of the slope) is the

$$\eta(t,x) = \int_{-\infty}^{\infty} i\omega A_0(\omega) J_0(2\omega\sqrt{x}) \exp(i\omega t) d\omega \tag{15}$$



where $J_0$ is Bessel function of the first kind of order zero. Note that we performed a linearization here by taking $\lambda = t$ and $\sigma = \sqrt{x}$, because we assume that in the deeper part of the domain, waves are small. The unknown coefficients $R(\omega)$ and $A_0(\omega)$ are to be determined from the continuity conditions at the toe of the slope. The zero subscript for $A$ is used because for *MODEL-1*, having a fully one-dimensional geometry, only one coefficient is needed. Later, other coefficients will also be needed when we will consider two spatial dimensions.

The linearized free surface and flux continuity conditions at the toe ($x = 1$) are given as

$$\varphi(x = 1^-) = \varphi(x = 1^+)$$
$$\partial_x \varphi(x = 1^-) = (D+1)\partial_x \varphi(x = 1^+) \tag{16}$$

which, in the matrix form, is given as

$$\begin{pmatrix} J_0(2\omega) & -1/i\omega \\ -\omega J_1(2\omega) & \sqrt{D+1} \end{pmatrix} \begin{pmatrix} A_0(\omega) \\ R(\omega) \end{pmatrix} = \begin{pmatrix} \tilde{\eta}^I(\omega)/i\omega \\ \sqrt{D+1}\,\tilde{\eta}^I(\omega) \end{pmatrix}. \tag{17}$$

In the system (17), the determinant of the matrix on the left does not become zero for any *real* frequency. However both $A_0$ and $R(\omega)$ have simple poles at $\omega = 0$ because the right-hand side diverges at $\omega = 0$. If one pursues, instead, a linear solution without resorting to the Carrier-Greenspan transformation, then the system (16) can be cast directly in terms of $\eta$ and $\partial_x \eta$ and then, the right-hand side of the resulting matrix system would be non-singular for *all real* frequencies. Such a system would be easily solvable by means of inverse Fourier transforms as shown by Ezersky et al. (2013b) who worked on a geometry featuring a sequence of slopes. Such a linear approach is useful but incapable of detecting the resonance phenomenon effectively because the solution is expressed in terms of an integration over a continuous spectrum of frequencies. In this work we will show, through a residue approach, that such a continuous integration of frequencies can be reduced into a summation over *discrete complex* frequencies, thereby enabling us to pinpoint resonance mechanism for *real* frequencies . Note that the conservation of energy requires that the amplitude of the incident wave is equal to that of the reflected wave with $|\tilde{\eta}^I(\omega)| = |R(\omega)|$ when the frequency is *real* . However this condition is relaxed when the frequency is no longer real because the energy density averaged over one cycle of oscillation evolves in time. For a discrete range of complex frequencies $\omega_k$ the determinant of the system in equation 17 vanishes. Therefore a non-trivial solution of (17) for $\omega = \omega_k$ and $\tilde{\eta}^I = 0$ can be found. These solutions are called free modes because they can be sustained without an incident wave and their energy originates solely from the initial conditions. Because of the radiation conditions, the reflected wave is a function of $t - x/\sqrt{D+1}$, thus it carries the energy in the offshore direction. In a different bathymetric setting (a circular submerged sill), a similar radiation problem has been studied by Longuet-Higgins (1967). Without the supply of energy from the incident wave the energy density must decay. The energy density being proportional to the square of the amplitude $|R(\omega)|^2 \exp(-2\Im\omega_k(t - x/\sqrt{D+1}))$, the imaginary part of the frequencies of the free modes must be positive in order to insure the decay. See Synolakis (1988) for a rigorous proof that the frequencies of the free modes are indeed on the upper complex half-plane for $D$ zero. Our argument based on the conservation of energy is more general and can be applied to any bathymetric profile.





We emphasize that the complex nature of these frequencies is not due to a dissipative process such as friction but it is a consequence of the decay of the energy density because of the transmission of the energy to the infinity. The total energy is preserved because both frequency $\omega_k$ and wave vector $\omega_k/\sqrt{D+1}$ are complex and they satisfy the usual non-dissipative dispersion relation $\omega = \sqrt{D+1}k$ where $k$ is the wavenumber.

Recently Stefanakis et al. (2015) attempted to calculate resonant frequencies for a geometry similar to Ezersky et al. (2013b) but imposed an offshore Dirichlet condition of the form $\eta = \sin(\omega t)$. With this boundary condition their results diverged at certain discrete *real* frequencies. They attributed this to resonance. However, those frequencies leading to divergence were real because the boundary condition they used did not allow radiation towards offshore. This corresponds to the non-physical case where the sum of the incident and reflected waves always remains equal to $\sin(\omega t)$. Hence the resonance they found has

limited physical meaning. In this work, we replace this approach with a physically realistic initial value problem. Not only that the resonant frequencies we will calculate will be complex, but also that their real parts will be substantially higher than those found by Stefanakis et al. (2015) when the discontinuity, $D$, is small (see Table 1). Only in the limit of large $D$, the complex roots of the determinant $(i\sqrt{D+1}J_0(2\omega) - J_1(2\omega))$ approach those of Dirichlet condition $J_0(2\omega) = 0$.

The residues theorem will clarify how these free modes will be excited by the incident wave. The solution for $A_0$ from (17)

is

$$A_0(\omega) = \frac{2\sqrt{D+1}}{\omega\left(i\sqrt{D+1}J_0(2\omega) - J_1(2\omega)\right)}\tilde{\eta}^I(\omega) \, . \tag{18}$$

Now consider an incident wave of the following form:

$$\eta_0^I(t,x) = \delta\left(t - t_0 + \frac{(x-1)}{\sqrt{D+1}}\right) \tag{19}$$

where $\delta$ is the Dirac's delta function. The zero index of $t$ in (19) relates to the phase of the incident wave. The Fourier transform

of (19), with respect to time, is then equal to $\exp(-i\omega t_0)/2\pi$ for $x = 1$. The response, $\varphi$, to such Dirac-type incident wave will be called Green's function $G(\lambda, t_0, \sigma)$. Remember that $A_0(\omega)J_0(2\omega\sigma)$ is the Fourier transform of $\varphi$. In (18) if we replace $\tilde{\eta}^I$ by $\exp(-i\omega t_0)/2\pi$, then $A_0(\omega)J_0(2\omega\sigma)$ will become the Fourier transform of the Green's function $\tilde{G}(\omega, t_0, \sigma)$. This Green's function in Fourier domain is given as

$$\tilde{G}(\omega, t_0, \sigma) = \frac{\sqrt{D+1}\exp(-i\omega t_0)}{\pi\omega\left(i\sqrt{D+1}J_0(2\omega) - J_1(2\omega)\right)}J_0(2\omega\sigma). \tag{20}$$

Any incident wave can be expressed in terms of a linear superposition of Dirac functions:

$$\eta^I(t,x) = \int_0^t \eta^I(t_0,x)\delta\left(t - t_0 + \frac{(x-1)}{\sqrt{D+1}}\right)dt_0 \, . \tag{21}$$





The response, $\varphi$, will then be

$$\varphi(\lambda,\sigma) = \int_0^{t(\lambda,\sigma=1)} G(\lambda,t_0,\sigma)\eta^I(t_0,x=1)dt_0 \ . \tag{22}$$

Because of the linearisation at the toe of the slope, the upper limit of the integration, $t(\lambda,\sigma=1)$, in (22) can be simply replaced by $\lambda$. We will use $\tilde{G}(\omega,t_0,\sigma)$ to recover $G(\lambda,t_0,\sigma)$. Accordingly the potential will become

$$\varphi(\lambda,\sigma) = \int_0^\lambda dt_0 \int_{-\infty}^\infty \tilde{G}(\omega,t_0,\sigma)\eta^I(t_0,x=1)\exp(i\omega(\lambda - t_0))d\omega \tag{23}$$

where the integration over the frequencies will be transformed to a series of residues. This is not a closed integral over the complex plane but a *real* line integral between $-\infty$ and $\infty$ except $\omega = 0$ which we circumvent with an infinitesimal semi-circle on the lower half-plane. The reason we use the lower half-plane is that we want the Green's function to vanish for negative values of $\lambda$. The whole integral can be cast into a closed integral by connecting $\infty$ to $-\infty$ along a semi-circle on the upper

complex plane and can be calculated using a residue summation. Here $\omega = 0$ is not the only pole. As a matter of fact $\tilde{G}(\omega,t_0,\sigma)$ has many poles in the upper half-plane (see equation 20). These poles are symmetrical with respect to the imaginary axis. The determinant of the matrix on the left-hand side of (17) becomes zero for these poles which are complex frequencies. When the determinant *and* the incident wave (right-hand side of (17)) are zero, one can still find non-trivial solutions for $A_0(\omega)$ and $R(\omega)$ and these correspond to free modes. Our aim is to understand the excitation of these free modes by the incident waves.

It is important to note that as $x \to \infty$ these free modes diverge. However this is not a problem if one wishes to find a solution in the coastal zone. We will expand the reflected wave in terms of these free modes. For the numerical calculation, the free mode expansion is truncated at a finite term $N$. This finite series also diverges for $x \to \infty$, thus the truncation error between the real reflected wave and the finite series expansion grows as $x$ increases but this is also not a problem if one wishes to find the wave field near the coast because the discrepancy (or error) propagates towards offshore.

Over the slope, the potential $\varphi$ is proportional to $J_0(2\omega\sigma)\exp(i\omega\lambda)$ (see (14)) and consequently $u$ and $\eta$ are off-phase by 90 degrees for real frequencies. The transmitted net power being proportional to $\eta u$ is then zero. However, as we explained above, the frequencies of the free modes, with net energy flux being towards offshore, must be complex. Thus $u$ and $\eta$ are not off-phase by 90 degrees over the slope. To perform the free mode expansion approach, let us rewrite the convolution (22) as

$$\varphi(\lambda(t,x),\sigma(t,x)) = \int_0^t dt_0 \int_{-\infty}^\infty \tilde{G}(\omega,t_0,\sigma)\eta^I(t_0,x=1)\exp(i\omega(t-t_0))d\omega \ , \tag{24}$$

thus we need to calculate the residues of the integral that we obtain if we substitute (22) into (24):

$$\varphi(\lambda(t,x),\sigma(t,x)) = \frac{1}{2\pi}\int_0^t dt_0 \int_{-\infty}^\infty \frac{2\sqrt{D+1}}{\omega\left(i\sqrt{D+1}J_0(2\omega) - J_1(2\omega)\right)} J_0(2\omega\sigma)\eta^I(t_0,x=1)$$

$$\times \exp(i\omega(t-t_0))d\omega \ . \tag{25}$$


**Table 1.** The complex natural frequencies multiplied by 2 using both Müller method and the asymptotic approach (see (A5)) for *MODEL-1* are tabulated. Note that $2\omega_k$ tend to the roots of $J_0$ for large $D$.

|  | $2\omega_1$ | $2\omega_2$ | $2\omega_3$ | $2\omega_4$ | $2\omega_5$ |
|---|---|---|---|---|---|
| (Müller) $D = 0.0$ | $2.98 + 1.28i$ | $6.17 + 1.61i$ | $9.34 + 1.81i$ | $12.4 + 1.96i$ | $15.6 + 2.07i$ |
| (Müller) $D = 1.0$ | $2.55 + 0.80i$ | $5.60 + 0.86i$ | $8.70 + 1.87i$ | $11.83 + 0.87i$ | $14.96 + 0.87i$ |
| (Asymp.) $D = 1.0$ | $2.52 + 0.70i$ | $5.56 + 0.70i$ | $8.68 + 0.70i$ | $11.81 + 0.70i$ | $14.94 + 0.70i$ |
| (Müller) $D = 5.0$ | $2.44 + 0.42i$ | $5.53 + 0.43i$ | $8.66 + 0.43i$ | $11.8 + 0.43i$ | $14.9 + 0.43i$ |
| (Asymp.) $D = 5.0$ | $2.43 + 0.40i$ | $5.53 + 0.40i$ | $8.66 + 0.40i$ | $11.8 + 0.40i$ | $14.9 + 0.40i$ |
| (Müller) $D = 20$ | $2.41 + 0.22i$ | $5.52 + 0.22i$ | $8.65 + 0.22i$ | $11.8 + 0.22i$ | $14.9 + 0.22i$ |

In an effort to calculate solitary wave runup, Synolakis (1987) evaluated a similar integral in his linear approach. However his integral was tasked to compute directly the free surface, $\eta$, rather than the potential and consequently did not have a singularity at $\omega = 0$. The Fourier transform of the particular solitary wave Synolakis (1987) considered approached zero so fast along

the infinite-radius integration contour that he was able to close his contour on the lower half-plane (upper half-plane in his convention). Thus his complex integration loop did not contain any of the complex frequencies we mentioned above. The only poles that remained within his closed contour were those of the Fourier transform of the solitary wave he considered and were, therefore, independent of the geometry. His technique is limited to this particular incident wave forcing. In his article, Synolakis (1987), did express the solution in terms of a summation over discrete frequencies but these frequencies can not be

interpreted as free mode frequencies not only because they are independent of the geometry but also because the result is only valid for times smaller than a critical time $t_c$. To see this, let us write down the integral (given as equation 2.6 in Synolakis (1987)) in our convention for $D = 0$:

$$\eta(t,x) = \frac{1}{2\pi} \int_{-\infty}^{\infty} \frac{2\Phi(\omega)J_0(2\omega\sqrt{x})}{(iJ_0(2\omega) - J_1(2\omega))} \exp(i\omega t) d\omega \tag{26}$$

where $\Phi$ is the Fourier transform of incident wave. In the integrant here, the exponential term $\exp(i\omega t)$ diverges on the lower

semi-circle. This divergence is counter-balanced by the Fourier transform of the incident wave for $t < t_c$ but not later. In order to calculate the residue summation corresponding to the integral in (25), we need to calculate the frequencies that make the denominator zero. Here, when $D \to \infty$, the approximate roots are given by the roots of $J_0(2\omega)$. These roots are real but when $D$ is large but finite, then the roots are complex and can be found using a perturbation approach given in the Appendix A. The perturbation approach is analytical but not as accurate as the numerical Müller scheme (see Press et al. (2007)). Note

that the Müller scheme requires an initial guess to calculate the complex root, we used the real roots of $J_0(2\omega)$ as the initial guesses. A comparison between these two approaches is displayed in Table-1. As seen in this table, as $D$ increases, radiation towards offshore becomes less efficient, therefore making the complex parts of the roots smaller. The real parts, also decrease





as $D$ increases. This is because the waves that propagate from the shore towards offshore create reflecting waves continuously because of the variable depth of the slope over which they are travelling. However, we have radiation damping in our case, some of the waves that reach the toe of the channel escape from the sloping part of the channel. In the absence of radiation damping, all waves reaching the toe of the channel would reflect back. These waves contribute to standing waves of low frequencies

over the sloping channel, because of the long distance they travel. Consequently, the relative weight of this low frequency component would increase if there is no or little radiation. In the case of any non-uniform bathymetry, at the high-frequency limit, ray theory can be used and the reflections will become minimal. This explains why the damping factor of the higher modes (imaginary parts of the eigenfrequencies) become larger. In the real geophysical settings where the discontinuities are less abrupt than the geometry depicted here (such as the edges of coastal shelves), the short waves will cross over the toes of

the slopes, essentially without reflection.

The geometry considered by Stefanakis et al. (2015) features two consecutive slopes. If one fixes the slope angles in their work to a single value, the resulting geometry would be the same as ours. The frequencies that they would have come up with in their solutions would have been the real roots of $J_0(2\omega)$. These frequencies are substantially smaller than the real parts of the frequencies we calculate (2.98 versus 2.4 for the fundamental mode) with radiation damping. We also investigated the case

with two consecutive slopes, but, unlike Stefanakis et al. (2015), we did include radiation damping. As an example case, we considered a geometry where the first slope is exactly the same as above, but at $x = 1$ this slope joins another, steeper, slope ($\beta$). This second slope reaches its end at $x = 1 + D/\beta$. In Figure (2), two of the lowest mode frequencies (real and complex ones, separately) are displayed as a function of $\beta$. These curves are obtained by finding the complex roots of the denominator of equation (15) of Ezersky et al. (2013b). It is seen in this figure that when $\beta$ is larger than about one, the imaginary parts of

the frequencies are decreasing functions of $\beta$. This is because for larger values of $\beta$, the radiation damping is less effective. Another feature is that the convergence to the asymptotic value of the real parts of the frequencies ($\beta$ leading to infinity) happens at smaller $\beta$ values for lower modes. The reason for this is that shorter wavelengths *feel* the second slope more.

Now let us return to the integral (25). This integral can be written completely in terms of $\lambda$ and $\sigma$ as:

$$\varphi(\lambda,\sigma) = \frac{1}{2\pi} \int\limits_0^{\lambda-2(1-\sigma)} dt_0 \int\limits_{-\infty}^{\infty} \frac{2\sqrt{D+1}J_0(2\omega\sigma)}{\omega\left(i\sqrt{D+1}J_0(2\omega) - J_1(2\omega)\right)} \eta^I(t_0,\sigma=1)$$

$$\times \exp(i\omega(\lambda-t_0))d\omega \tag{27}$$





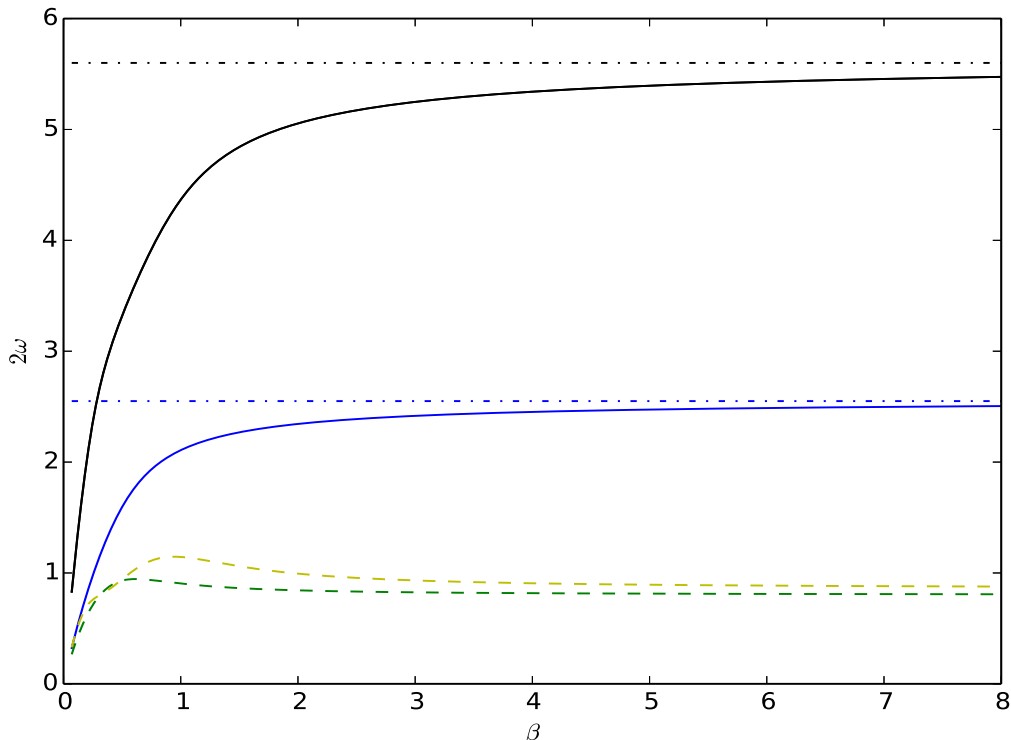

**Figure 2.** The blue and black continuous curves are the real parts of twice the first and second mode frequencies, respectively, for the two-slope case. The flat, dash-dotted lines are the asymptotic values for $\beta$ leading to infinity for these two modes. The green and yellow dashed curves are the imaginary parts of twice the first and second mode frequencies. This figure corresponds to the specific case in which the second slope reaches its end at $x = 1 + D/\beta$ where $D = 1$.

for $\lambda > 2(1-\sigma)$ because of the causality ($\varphi$ becomes zero otherwise, because the disturbance takes $2(1-\sigma)$ to travel from the toe of the slope to the point $\sigma$). A residue summation transforms (27) into:

$$\varphi = \left( 2 \int_0^{\lambda-2(1-\sigma)} dt_0\, \eta^I(t_0, x=1) \right) + \sum_{k=-\infty, k\neq 0}^{+\infty} \frac{i\sqrt{D+1}J_0(2\omega_k\sigma)}{\omega_k\left(i\sqrt{D+1}J_0'(2\omega_k) - J_1'(2\omega_k)\right)}$$

$$\times \int_0^{\lambda-2(1-\sigma)} dt_0\, \eta^I(t_0, x=1)\exp\left(i\omega_k(\lambda - t_0)\right) \tag{28}$$

where $J_0'$ and $J_1'$ are the derivatives of $J_0(2\omega_k)$ and $J_1(2\omega_k)$ with respect to $2\omega_k$. The first integral in (28) comes from the residue associated with the pole at $\omega = 0$. It is calculated by using the fact that $J_0(0) = 1$ and $J_1(0) = 0$. Using the Leibnitz



rule, the partial derivatives of the potential are then

$$\partial_\lambda \varphi = 2\eta^I(\lambda - 2(1-\sigma), x = 1) + \sum_{k=-\infty, k\neq 0}^{\infty} \frac{i\sqrt{D+1}J_0(2\omega\sigma)}{\omega_k\left(i\sqrt{D+1}J_0'(2\omega_k) - J_1'(2\omega_k)\right)}$$
$$\times \eta^I(\lambda - 2(1-\sigma), x = 1)\exp\left(2i\omega_k(1-\sigma)\right)$$
$$+ \sum_{k=-\infty, k\neq 0}^{\infty} \frac{-\sqrt{D+1}J_0(2\omega_k\sigma)}{i\sqrt{D+1}J_0'(2\omega_k) - J_1'((2\omega_k)} \int_0^{\lambda-2(1-\sigma)} dt_0\, \eta^I(t_0, x = 1)\exp\left(i\omega_k(\lambda - t_0)\right) \tag{29}$$

and

$$\partial_\sigma \varphi = 4\eta^I(\lambda - 2(1-\sigma), x = 1) + \sum_{k=-\infty, k\neq 0}^{\infty} \frac{2i\sqrt{D+1}J_0(2\omega_k\sigma)}{\omega_k\left(i\sqrt{D+1}J_0'(2\omega_k) - J_1'(2\omega_k)\right)}$$
$$\times \exp(i\omega_k 2(1-\sigma))\eta^I(\lambda - 2(1-\sigma), x = 1)$$
$$+ \sum_{k=-\infty, k\neq 0}^{\infty} \frac{-2i\sqrt{D+1}J_1(2\omega_k\sigma)}{i\sqrt{D+1}J_0'(2\omega_k) - J_1'(2\omega_k)} \int_0^{\lambda-2(1-\sigma)} dt_0\, \exp(i\omega_k(\lambda - t_0))\eta^I(t_0, x = 1). \tag{30}$$

The equation given in (9) is, in essence, the linear wave equation with cylindrical symmetry. Because of this, the partial derivative of its regular solution with respect to $\sigma$ must be zero at $\sigma = 0$. A quick inspection of (30) reveals that the terms in the first and second line are not equal to zero individually, and their collective sum will involve a truncation error. This truncation error is subject to an amplification in the estimation of $\partial_\sigma(\varphi)/\sigma$ which is used to calculate both $\eta$ and $u$. To remedy this, we use the fact that, near $\sigma = 0$, the value of $\partial_\sigma\varphi/\sigma$ is approximately equal to $2\partial_{\lambda\lambda}^2\varphi$ due to L'Hôspital's rule. Hence, we use the numerical derivative of (29) to find the non-linear contribution to runup.

## 4 Resonance sensitivity for *MODEL-1*

In this section we consider a monochromatic incident wave of type $\eta_0^I(\omega)\sin(\omega(t+(x-1)/\sqrt{D+1}))\theta(t+(x-1)/\sqrt{D+1})$. We will study the evolution in the large time limit, so-called the steady-state regime. The transient regime will be discussed in the next subsection. Here $\theta$ is the Heaviside function that takes value 1 for positive argument and 0 otherwise. The resulting wave on the slope in the linear approximation will then

$$\eta(t,x) = \left(A_0(\omega)\frac{\exp(i\omega t)}{2} + A_0(-\omega)\frac{\exp(-i\omega t)}{2}\right)\omega J_0(2\omega\sqrt{x}) \tag{31}$$

for $t \to \infty$. This equation follows naturally from equation (15) where instead of evaluating the integral, we add the two contributions coming from $\omega$ and $-\omega$. Taking into account that $J_0(0) = 1$ the associated runup, $r = \eta(t, \sigma = 0)$, becomes

$$\omega|A_0(\omega)|\cos(\omega t + \phi_A) \tag{32}$$

in the linearised theory. Here $\phi_A$ is the argument of complex number $A_0(\omega)$. In a real situation that would occur in the nature where the monochromatic incident wave gets generated at a particular instant $t = 0$, the expression

$$\omega|A_0(\omega)| \tag{33}$$





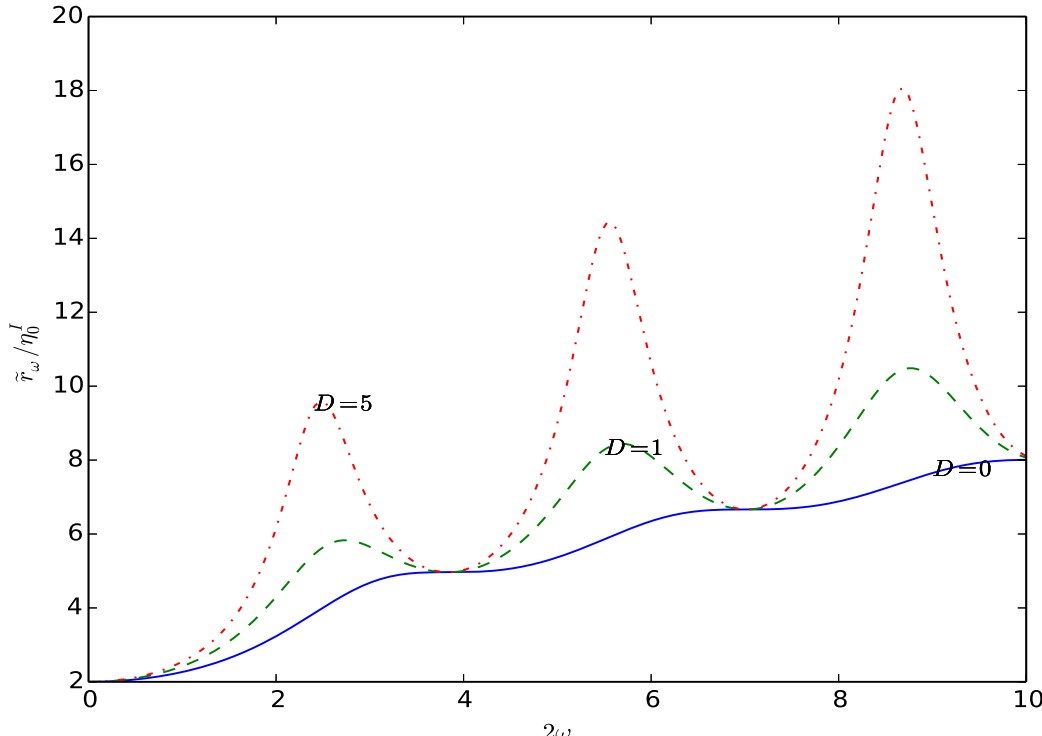

**Figure 3.** The limiting amplitude of runup, $\tilde{r}_\omega$, normalised to the amplitude of the incident wave is shown as a function of the frequency of the incident wave multiplied by 2 for *MODEL-1*. The depth discontinuity $D$ is 0 for the blue continuous curve, 1 for the green dashed curve, and 5 for the red dot-dashed curve.

in (32) provides a limiting value of amplitude of oscillation of the runup. This limiting amplitude, normalised to the amplitude of the incident wave, is displayed in Figure (3) as a function of $2\omega$ where $\omega$ is the frequency of the incident wave. In this figure local maxima of the limiting amplitude of the runup can be observed for $D = 1$ and $D = 5$ but not for $D = 0$ where this amplitude steadily increases with the frequency of the incident wave. Therefore there is no resonance for $D = 0$.

5     In the limit of large $D$, the local maxima of the limiting amplitude of the runup occur at the frequencies where $J_0(2\omega) = 0$ and the value of the limiting amplitude of the oscillation of the runup at those maxima is given as

$$2\sqrt{D+1}|\eta_0^I(2\omega_k)/J_1(2\omega_k)| \qquad (34)$$

where $2\omega_k$ is the $k^{\text{th}}$ root of Bessel function $J_0$. Hence, the runup sensitivity to $\omega$ increases as $D$ increases. For high frequencies, the asymptotic approximation, $\sqrt{\frac{1}{\pi\omega}}\cos(2\omega - 3\pi/4)$, can be used for $J_1(2\omega)$ and any consecutive local maxima and minima

10  of $\omega|A_0(\omega)|$ are then separated by $\pi/4$. In this high frequency and large $D$ limit, the amplitude of runup increases by

$$2(\sqrt{D+1} - 1)\sqrt{\pi\omega}|\eta_0^I| \qquad (35)$$




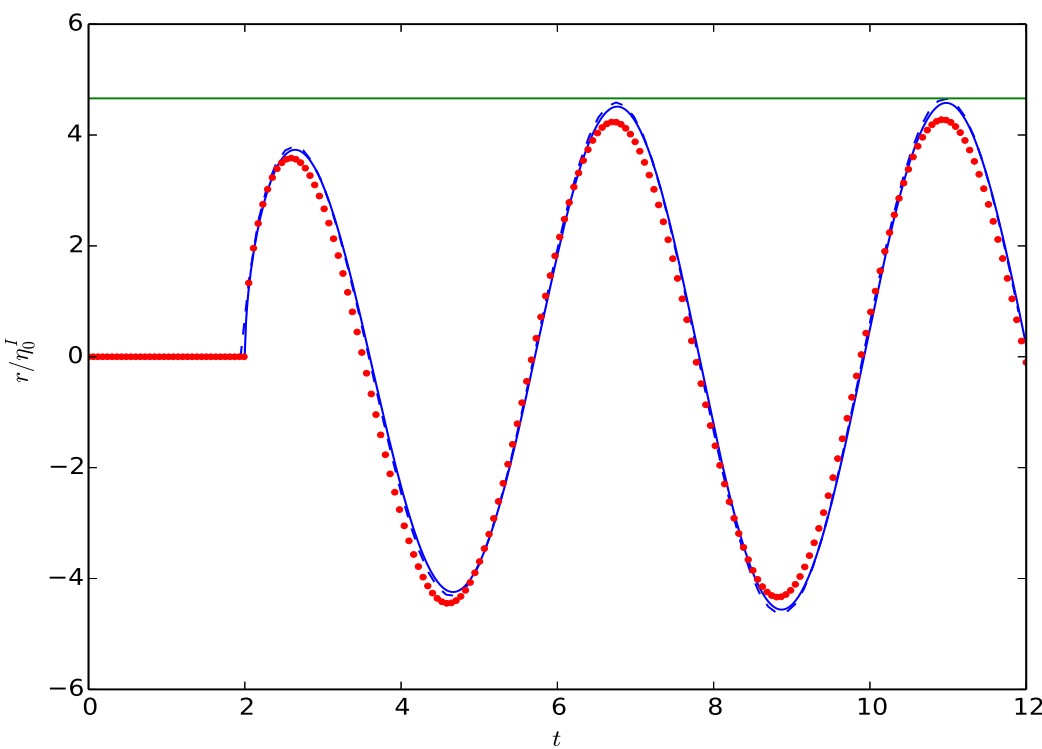

**Figure 4.** Continuous and dashed '-.' blue curves display the runup normalised to the amplitude of incident wave for *Model I* with $D = 0$. The frequency of the incident wave is equal to the real part of the first-mode resonant frequency $\omega_1$ (see Table 1). The continuous blue curve is obtained from the series of residues (see equation 29). The dashed blue curve is the runup calculated using the Fast Fourier Transform approach. The green horizontal line on the top part of the figure is the limiting amplitude given by $\omega^I |A_0(\omega^I)|/\eta_0^I$ where $\omega^I$ is the frequency of the incident wave. The red bullets are the runup produced by a wave-maker on an *infinite* constant slope. The action of the wave-maker is represented by the "tsunamigenic" seafloor motion given by $h(t, x) = -2\eta_0^I \delta(x-1)\cos(\Re\omega_1 t)\theta(t)/\Re\omega_1$ where $\theta$ is the Heaviside function, $h$ is the seafloor uplift and $\Re$ denotes the real part.



as $\omega$ moves from the local minimum to the next maximum.

These results indicate that in the real world where $D$ is often much smaller than one, the resonance is an insignificant phenomenon. For two-slope cases the resonance is even less significant because the imaginary parts of eigenfrequencies are decreasing functions of second slope, $\beta$.

One last remark relates to the power laws for runup, provided by Didenkulova et al. (2009). The Figure (3) shows that, when $D$ is large, the denominator of 18 has roots that are closer to the real axis. This essentially means that for large $D$ explicit relations for power laws might not be possible to derive.

## 4.1   Transient regime

The resonant phenomena we discussed above do not set in immediately upon the entrance of the incident wave into the

slope region. It is important to know how fast the limiting amplitude of oscillation of the runup will be reached, because in a real situation the incident wave will have a finite duration, a fact that was not taken into account in the large time limit analysis. For that purpose the residue series in equation (29) must be evaluated. Here, the incident wave is, again, taken as $\eta_0^I \sin(\omega(t+(x-1)/\sqrt{D+1}))\theta(t+(x-1)/\sqrt{D+1})$. The resulting runup is displayed in Figure (4). A good agreement between runup obtained from residues summation and the fast Fourier transform approach (Appendix B) can be seen in this

figure (continuous and dashed curves respectively). The limiting amplitude for the runup is almost reached after just one oscillation (see the horizontal limiting line on the top part of the figure) In Figure 5 the runup is displayed as function of time, for incident waves of three different frequencies. In this figure no significant change in runup can be observed as the frequency of the incident wave deviates from the resonant frequency . This us due to the fact that the depth discontinuity $D$ is zero. On the other hand in Figure(6) where $D$ is equal to 5, the runup increases as the incident wave frequency approaches the resonant

frequency. A comparison between Figures (5) and (6) reveals that in the second figure it takes longer for the runup to reach its limiting value. This is due to the fact that for larger $D$ the imaginary part of the frequency of the free mode is smaller. This means that if the incident wave excitation were to be cut at a given time, the standing wave oscillations over the slope would last longer (before eventually decaying due to the radiation) when $D$ is larger. Similarly, for large $D$, it takes also longer for the standing wave regime to reach its limiting amplitude. It is also important to note that in both Figures (5) and (6) the initial time

derivative of the runup is very large, leading to the glitch at $t = 6$ in Figure (6). No such glitch exists in Figure (5) because there is minor reflection from the toe. In the real nature, waves do not "switch on" at a given time, at once. Therefore their initial profiles do not have discontinuous spatial derivatives. As a matter of fact, such discontinuities will trigger short-frequency waves which can not be properly modeled using shallow water approach anyway. The discontinuity mentioned here is seen much more clearly in the shoreline velocity field in Figure (7).

## 30  4.2   Nonlinear effects

Several features of the runup phenomenon can be considered nonlinear. One of them is the fact that the incident wave offshore might be large and the linearization there may be completely invalid. Such a linearization is very common in analytical studies, however there are exceptions such as Kânoğlu and Synokalis (2005). The nonlinear effects increase as the waves become





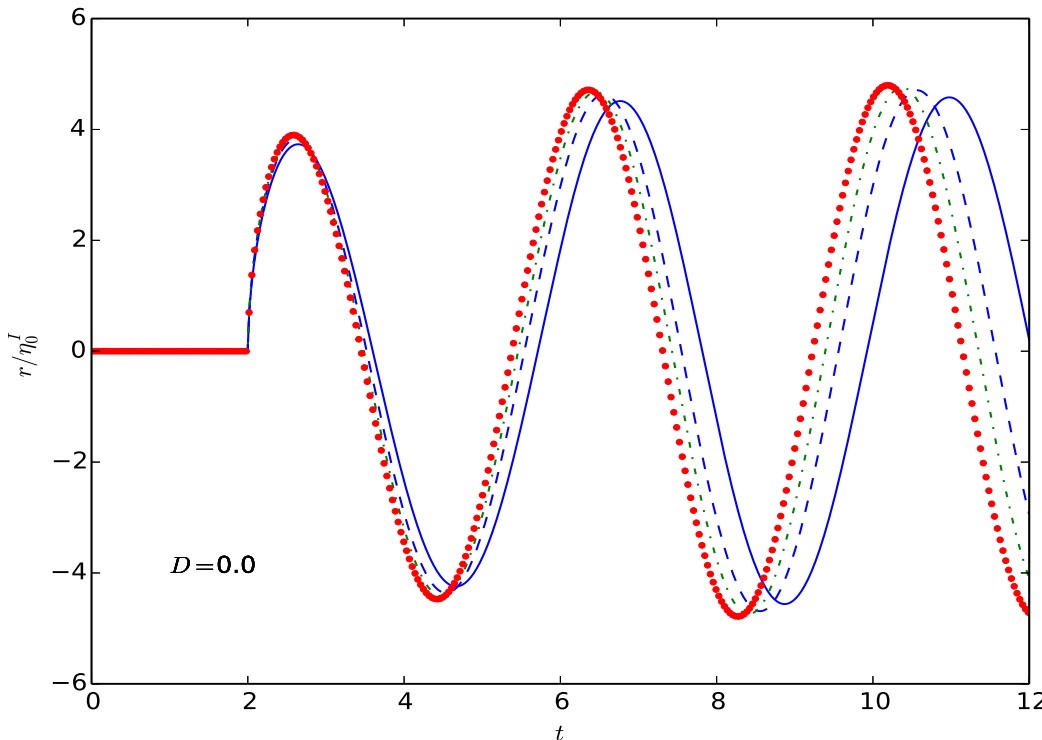

**Figure 5.** Runup normalised to the amplitude of the incident wave for *MODEL-1*. The frequency of the incident wave is $\Re\omega_1$ for the continuous curve, $1.05\Re\omega_1$ for '- -', $1.075\Re\omega_1$ for '-.', and $1.1\Re\omega_1$ for the red bullets. Depth discontinuity $D$ is zero. Refer to the first row of table 1 for the value of $\omega_1$.

higher towards the coast, however, as indicated by Pelinovsky and Mazova (1992), these near-shore nonlinear effects do not affect the maximum shoreline velocity, they do, on the other hand, affect the timing of the maximum. In the $\lambda$-domain, the difference between the linear runup and nonlinear runup, assuming that the waves do not break, is a function of $u^2$. This means that, the only correction we have to take into account to calculate the nonlinear runup is $u_s^2/2$ at any *time* $\lambda$. Figure (8) shows

5 the difference between linear and nonlinear runup, and the time as a function of $\lambda$ respectively. Another important point to mention is that for the mapping between $t$ and $\lambda$ to be one-to-one, therefore invertible, $\partial u/\partial\lambda$ must be strictly less than one. This condition is violated if we do not apply a smoothing filter to the incident wave (even for infinitesimally small amplitudes), for the transient case. This violation is equivalent to a large acceleration of the shoreline position and would cause the non-dimensional breaking parameter "Br" defined in equation (11) in Pelinovsky and Mazova (1992) to eventually become infinite.

10 The offshore boundary condition studied in the numerical work of Stefanakis et al. (2015) would have led to infinite amplitude of the runup at their real resonant frequencies. Their fully nonlinear numerical solutions, however, converge to finite limits





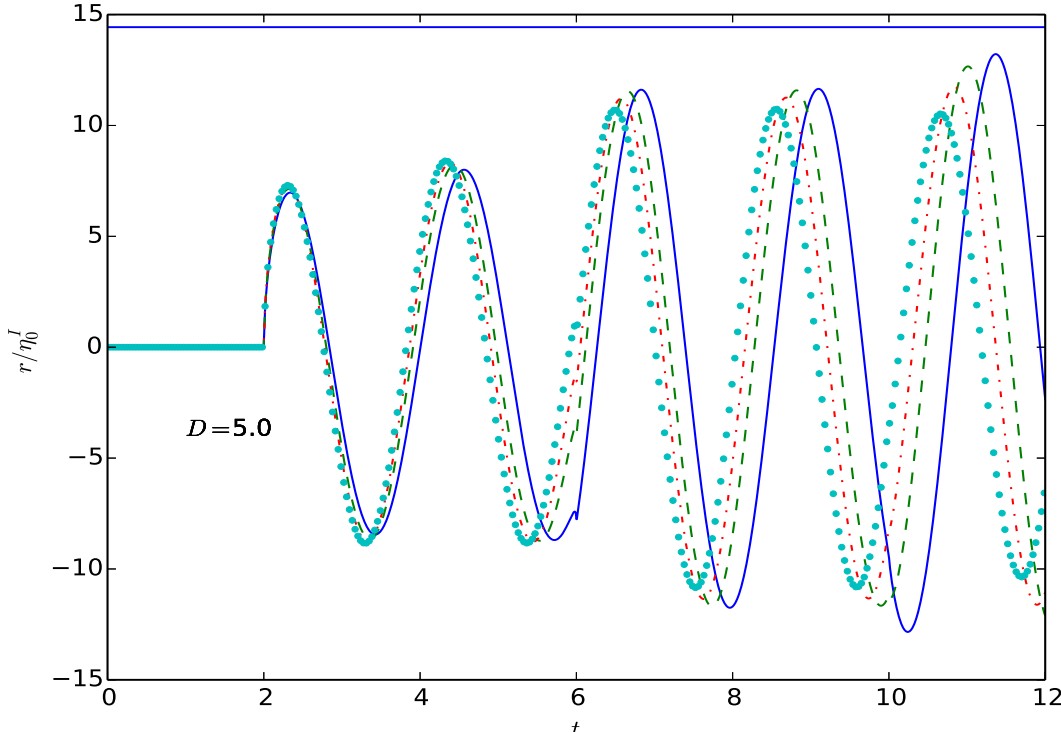

**Figure 6.** Runup normalised to the amplitude of the incident wave for *MODEL-1*. The frequency of the incident wave is $\Re\omega_1$ for the continuous curve, $1.05\Re\omega_2$ for dashed curve, $1.075\Re\omega_2$ for dashed-dotted curve, and $1.1\Re\omega_2$ for dotted curve. The flat horizontal line above is the limiting amplitude for $\omega = \Re\omega_1$. Depth discontinuity $D$ is 5. Refer to the first row of Table 1 for the value of $\omega_1$. Note that the wave arrives at the shore at $t = 2$ and the first reflection from the toe of the slope reaches the shore at $t = 6$ for which the continuous plot includes a slight glitch. In the main text we elaborate on this glitch.

for these frequencies due to nonlinearities. The reason the nonlinearities yield such a consequence is because the nonlinear interaction of the modes imply amplitude-dependent natural frequencies.

# 5   Resonance for infinite slope

In this section we investigate the resonant frequencies of the waves produced by a wave-maker placed on an infinite, constant
5   slope. In the analysis we will allow the waves to progress in the offshore direction, unrestricted. When the the wavelength produced by the wave-maker matches the distance of the wave-maker to the shoreline on might expect a resonance to occur. To see if this is really the case, let us go into a little further detail about the nature of the wave-maker.





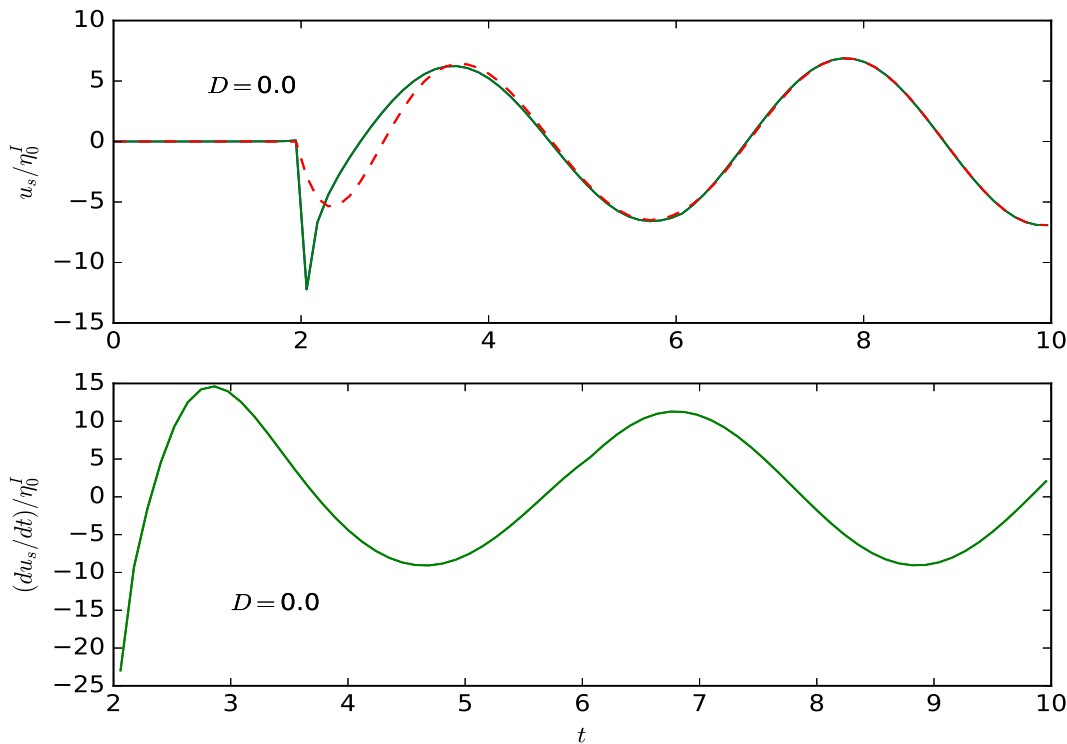

**Figure 7.** The top figure is the shoreline velocity $u_s$ normalised to the amplitude of the incident wave as a function of $t$ (*Model 1*). The frequency of the incident wave is $\Re\omega_1$. The continuous green curve is the shoreline velocity corresponding to incident wave $\eta = \eta_0^I \sin(\omega(t + (x-1)/\sqrt{D+1}))\theta(t + (x-1)/\sqrt{D+1})$. The red, dashed curve is the same for the incident wave that has been smoothed by multiplying it by $\tanh(t + (x-1)/\sqrt{D+1})$. The bottom figure is the time derivative of the shoreline velocity for the smoothed incident wave.

First of all, a hypothetical wave-maker will invariably produce waves of equal amplitude propagating in both directions. This is why, we will always have a factor 2 in front of the wave-maker amplitudes in the formulation. The wave-maker will be represented by a time-periodic uplift and subsidence of the seafloor. The displacement of the seafloor will then be given by $h(t,x) = 2\tilde{\eta}_0(\omega)\delta(x - x_0)\exp(i\omega t)/i\omega$ where $x_0$ is the distance of the wave-maker to the shore. This wave-maker will inject/suck fluid into the system (and from the system) at a rate $\int_0^\infty dx\partial_t h = 2\tilde{\eta}_0 \exp(i\omega t)$. Since the wave-maker is localised at $x = x_0$ the jump in the flux must be equal to the rate of the fluid injected/sucked, that is:

$$x_0\left(u(t, x_0^+) - u(t, x_0^-)\right) = 2\tilde{\eta}_0(\omega)\exp(i\omega t) \tag{36}$$

where $x_0$ is both the horizontal position of the source and the non-dimensional depth at that position. Furthermore, the continuity of the free surface at $x = x_0$ requires that

$$\eta(t, x^+) = \eta(t, x^-). \tag{37}$$





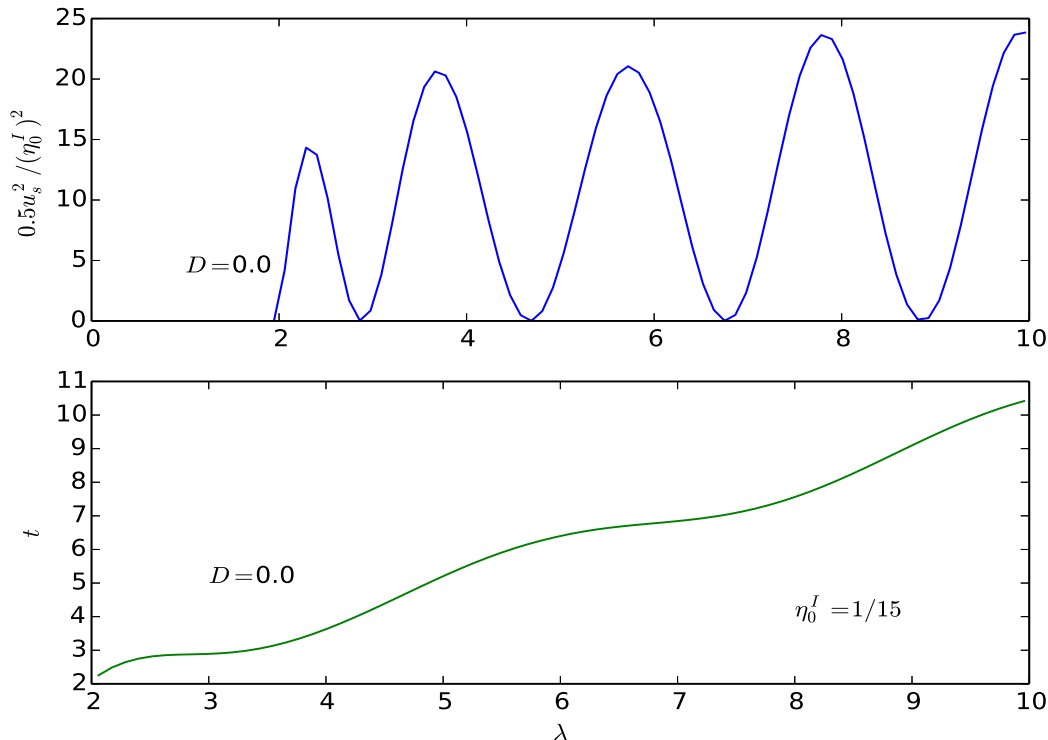

**Figure 8.** The top figure is the is the difference between the linear and non-linear run-up normalised to the the square of the amplitude of the incident wave (*Model 1*) as a function of $\lambda$. The frequency of the incident wave is $\Re\omega_1$. The bottom figure is the time as the function of $\lambda$, corresponding to incident wave, of amplitude $1/15$. Both figures assume an incident wave smoothed by the same $tanh$ factor as in the previous figure.

The causality principle requires that the waves must progress in the $+x$ direction for $x > x_0$. For $\omega > 0$ a solution that satisfies this radiation condition is given by $H_0^{(2)}(2\omega\sqrt{x})\exp(i\omega t)$ where $H_0^{(2)}$ is the Hankel function of the second kind, of order zero. A linear solution of the wave equation that satisfies the boundary conditions in equations (36) and (37) is, for $\omega > 0$,:

$$\eta(t,x) = \begin{cases} 2i\pi\tilde{\eta}_0(\omega)J_0(2\omega\sqrt{x_0})H_0^{(2)}(2\omega\sqrt{x})\exp(i\omega t) \text{ for } x > x_0 \\ 2i\pi\tilde{\eta}_0(\omega)H_0^{(2)}(2\omega\sqrt{x_0})J_0(2\omega\sqrt{x})\exp(i\omega t) \text{ for } x < x_0 \end{cases}. \tag{38}$$

5   When the frequency of wave-maker is negative it suffices to replace (38) by its complex conjugate. The solution given by (38) is the response of the system to a time-periodic injection/suction action. It is, in essence, a spatial Green's function in the frequency domain, multiplied by $\tilde{\eta}_0(\omega)$. The runup resulting from the sea-bed displacement $h(t,x) = 2\tilde{\eta}_0(\omega)\delta(x -$





$x_0) \exp(i\omega t)/i\omega$ is then

$$r = 2i\pi\tilde{\eta}_0(\omega)H_0^{(2)}(2\omega\sqrt{x_0})\exp(i\omega t). \tag{39}$$

This is a steady-state solution but we can use this solution to obtain the *transient* runup response, again via (38) by replacing $\tilde{\eta}_0(\omega)$ by the temporal Fourier transform of $\dfrac{\partial_t h}{\delta(x - x_0)}$. To summarize this process, let us write the seafloor displacement, in shorthand, as

$$h(t,x) = h^*(t)\delta(x - x_0). \tag{40}$$

Accordingly, we obtain

$$\tilde{\eta}(\omega) = \frac{1}{2\pi}\int\limits_{-\infty}^{\infty}\partial_t h^* \exp(-i\omega t)d\omega. \tag{41}$$

The inverse temporal Fourier transform of $\tilde{\eta}(\omega)$ gives us the free surface evolution.

As an application let us consider a seafloor velocity model as

$$\partial_t h = 2\eta_0^I \sin(\Re(\omega_1)t)\theta(t)\delta(x - 1) \tag{42}$$

which corresponds to a source with sinusoidal time dependence acting at $x = 1$. The resulting runup is displayed in Figure (4) as red dots where is is plotted together with the incident wave solutions for the geometry of *MODEL 1* with $D = 0$. It is interesting to observe that the runup for infinite slope with transient wave-maker slightly undershoots the runup for the *MODEL 1* in the steady-state limit. This is due to the lack of reflections by the toe in the infinite slope case.

## 6   Normally incident wave from a 2D ocean into a bay

Analytical studies that consider runup in 2D are rare. There are, however, some studies that combine 1D analytical approaches with 2D numerical simulations such as Choi et al. (2011). In this section we shall study the *MODEL-2*. We consider an incident wave of the form $\eta_0^I(\omega)\exp(i\omega(t + (x - 1)/\sqrt{D + 1}))$ coming from the open ocean into a sloping bay of width $2a$. This wave will be reflected back by the the shallower part of the bay. This reflected wave will then be subjected to geometrical spreading in the open ocean. Consequently, the one-dimensional nature of the waves will be lost in the deeper part of the bay. We will now derive a simple mathematical formulation to model these phenomena.

An approximate, low-frequency, two-dimensional solution of the linear wave equation in the deeper part of the channel can be given as

$$\varphi = \exp(i\omega t)\exp\left(\sqrt{(n\pi)^2/a^2 - \omega^2}(x - 1)\right)\cos(n\pi(y + a)/a) \tag{43}$$

for $\omega << \pi/a$. Here the $y$ axis is perpendicular to the normal incidence (see figure 1). This solution is only valid for constant unit depth but this is not a problem because it decays quickly for decreasing $x$, towards the shore. For integer and half-integer


values of $n$, the potential, $\varphi$, given by equation (43) satifies the non-flux condition at the side walls of the channel ($\partial_y\varphi = 0$ at $y = \pm a$). We wish the the solution to be symmetrical about the $x$ axis, this automatically excludes the half-integer values of $n$. This constant depth solution has been extended to the case of constant slope by Mei et al. (2004) and Zhang and Wu (1999) using the confluent hypergeometric function $M$. Their solution reads as

$$\exp\left(-\frac{\pi n x}{a} + i\omega t\right) M\left(-\frac{\omega^2 a}{2n\pi} + \frac{1}{2}, 1, 2\pi n x/a\right) \cos\left(n\pi \frac{y+a}{a}\right). \tag{44}$$

The full solution for the deeper part of the channel can then be expressed in terms of a linear combination of (44) and the linear, one dimensional solution over the slope:

$$\varphi_\omega(t,x) = A_0(\omega) J_0(2\omega\sqrt{x}) \exp(i\omega t) +$$

$$\sum_{n=1}^{\infty} A_n(\omega) \exp\left(-\frac{\pi n x}{a}\right) M\left(-\frac{\omega^2 a}{2n\pi} + \frac{1}{2}, 1, 2\pi n x/a\right) \cos\left(n\pi \frac{y+a}{a}\right) \exp(i\omega t) \tag{45}$$

where the unknown coefficients $A_0, A_1, ..$ are to be determined from the boundary conditions at the mouth of the channel

The bulk of the incident wave will be reflected back by the solid boundary at $x = 1, |y| > a$. Consequently the wave in the open ocean will be $\eta = 2\eta_0^I \cos(\omega(t - (x-1)/\sqrt{D+1}))$ perturbed by the waves radiating from the mouth of the channel and the scattering from channel mouth corners. The potential in the open sea then reads

$$\varphi_\omega(t, x > 1, y) = 2\frac{\eta_0^I(\omega) \cos\left(\omega(x-1)/\sqrt{D+1}\right)}{i\omega} \exp(i\omega t)$$

$$+ \int_{-a}^{a} d\tilde{y} \frac{S(\omega,\tilde{y})}{-2i(D+1)} H_0^{(2)}\left(\frac{\omega}{\sqrt{D+1}} |(x-1)\hat{\mathbf{i}} + (y-\tilde{y})\hat{\mathbf{j}}|\right) \exp(i\omega t) \tag{46}$$

where $\hat{\mathbf{i}}$ and $\hat{\mathbf{j}}$ are unit vectors in $x$ and $y$ directions respectively. Note that the Hankel function $H_0^{(2)}$ satisfies both the wave equation in two dimensions and the radiation condition for $\omega > 0$. For $\omega < 0$, on the other hand, Hankel function will be replaced by its complex conjugate alongside the rest of the terms. The integral in equation (46) represents the potential of the waves radiating from the mouth of the channel. Function $S(\omega,\tilde{y})$ is the unknown virtual sources distribution along the mouth of the channel. This source distribution will be determined by matching the potentials given in equations (45) and (46).

Let us now consider the fluid flow corresponding to a source distribution ,$S(\omega,\tilde{y})$, along the width element $\delta\tilde{y}$, placed at position $(1,\tilde{y})$. A logarithmic approximation can be used for the Hankel function for small arguments ($H_0^{(2)}(z) \approx (-2i/\pi)\log(z)$). Considering also that the fluid velocity $\mathbf{v}$ is equal to the gradient of the potential, the velocity vector due to sources in the element, $\delta y$, becomes

$$\mathbf{v} = \frac{\delta\tilde{y} S(\omega,\tilde{y})}{\pi(D+1)} \hat{\mathbf{d}}/d \tag{47}$$

for a near-field target position at distance $d$ from the source ($\hat{\mathbf{d}}$ is the unit vector pointing from the source towards the target). The factor $\pi$ in the denominator means that the offshore direction flux due to this source element is equal to $\delta\tilde{y}S$. Note that as $x \to 1^+$, the projection of $\mathbf{v}$ on the $x$ axis becomes zero for $|y - \tilde{y}| > |\delta\tilde{y}|$. Thus $v_x$ is zero almost everywhere but its integral in



the $y$-direction is finite and is equal to $\delta \tilde{y} S / (D+1)$, meaning that the $x$ component of the velocity due to this source is Dirac's delta function in the limit $x \to 1^+$ . To match the depth-integrated $v_x$ along the mouth we need to satisfy

$$(D+1)\frac{\partial \varphi_\omega(x,y)}{\partial x}\Big|_{x=1^+} = S(\omega,y) = \frac{\partial \varphi_\omega(x,y)}{\partial x}\Big|_{x=1^-}. \tag{48}$$

The second condition to be satisfied is the continuity of $\eta$ itself, across the mouth. This condition reads

$$\varphi_\omega(x,y)|_{x=1^+} = \varphi_\omega(x,y)|_{x=1^-}. \tag{49}$$

Here it is important to note that the condition (49) is an integral equation for the source distribution. The condition (48), on the other hand, is *not* an integral equation because the field inside the channel is governed using summations rather than an integral. In order to solve a system that simultaneously involves an integral equation and a set of algebraic equations we shall expand the source distribution, $S(\omega,y)$, in terms of even-order Legendre polynomials as

$$S(\omega,y) = \sum_{n=0}^{N-1} \tilde{S}_n(\omega) P_{2n}\left(\frac{y}{a}\right) \tag{50}$$

where the even indices of the Legendre polynomial were used to make sure that the solution is symmetric with respect to the symmetry axis of the sloping channel ($y=0$). The coefficients $S_n(\omega)$ in (50) and $A_0, A_1, ..., A_{N-1}$ in (45) (we truncate the series in (45) to $N-1$) are solved in such a way to minimize the following penalty integral along the mouth:

$$\int_{-a}^{a} \left( \left| \frac{\partial \varphi_\omega(x=1^-,y)}{\partial x} - S(\omega,y) \right|^2 + \left| \varphi_\omega(x=1^-,y) - \varphi_\omega(x=1^+,y) \right|^2 \right) dy. \tag{51}$$

Thus we have $2N$ unknowns. The integral (51) is evaluated numerically using Gauss quadrature. For precision, the number of quadrature points we use is larger than $2N$. The equations (48) and (49) have to be satisfied at all quadrature points, making the resulting system over-determined. We use weighted least-squares approach to solve this over-determined system. Algebraically speaking, this simply corresponds to writing down equations (48) and (49) for each quadrature point and multiplying each equation by the square-root of the corresponding quadrature weight and solving the resulting linear, over-determined system by using conventional least-squares method.

Note that in equations (45) and (50), the only terms that are responsible for the *net* flux from the channel to the open sea are $A_0$ and $\tilde{S}_0$ since the following integrals

$$\int_{-a}^{a} \cos\left(n\pi \frac{y+a}{a}\right) dy$$

$$\int_{-a}^{a} P_{2n}(y/a) dy \tag{52}$$

vanish for $n \neq 0$. The conservation of the net flux requires that $-2a\omega A_0 J_1(2\omega) = 2a\tilde{S}_0$ where $J_1$ is Bessel function of order 1. When the frequency of the incident wave is equal to that of the free oscillation of the system, the coefficients $A_0$ and $\tilde{S}_0$



**Table 2.** The complex natural frequencies multiplied by 2 are tabulated for *Model* 2. These are essentially those $2\omega_k$ that are the roots of $1/A_0(\omega)$. In this table those lines with "(int)" list the frequencies calculated using $A_0(\omega)$ obtained from minimising the penalty integral (51). The lines with "(conf)", on the other hand, use (59) from the conformal mapping formulation for this quantity. Note that $2\omega_k$ tend to the roots of $J_0$ for large $D$ or small $a$.

| | $2\omega_1$ | $2\omega_2$ | $2\omega_3$ | $2\omega_4$ | $2\omega_5$ |
|---|---|---|---|---|---|
| $a = 0.05, D = 0$ (int) | 2.26 + 0.056i | 5.27 + 0.126i | 8.32 + 0.19i | 11.4 + 0.25i | 14.5 + 0.31i |
| $a = 0.05, D = 0$ (conf) | 2.26+0.056i | 5.27+0.126i | 8.32+0.192i | 11.38+0.253i | 14.46+0.310i |
| $a = 0.1, D = 0$ (int) | 2.18 + 0.11i | 5.14 + 0.23i | 8.16 + 0.34i | 11.2 + 0.44i | 14.3 + 0.52i |
| $a = 0.1, D = 0$ (conf) | 2.18+0.107i | 5.13+0.234i | 8.14+0.346i | 11.17+0.446i | 14.20+0.533i |
| $a = 0.1, D = 1$ (int) | 2.28 + 0.057i | 5.30 + 0.13i | 8.37 + 0.20i | 11.45 + 0.26i | 14.6 + 0.32i |
| $a = 0.1, D = 5$ (int) | 2.35 + 0.020i | 5.43 + 0.045i | 8.53 + 0.07i | 11.65 + 0.095 | 14.8 + 0.12i |
| $a = 0.2, D = 0$ (int) | 2.05 + 0.197i | 4.95 + 0.41i | 7.95 + 0.58i | 11.0 + 0.74i | 14.0 + 0.88i |
| $a = 0.2, D = 0$ (conf) | 2.05+0.197i | 4.92+0.410i | 7.84+0.572i | 10.75+0.661i | 16.6+0.61i |

should diverge. In order to determine the frequencies of these free mode oscillations (natural frequencies), the roots of $1/A_0$ are sought in the complex plane using the Müller method.

A quick look at Table 2 reveals that for any mode, with the decreasing channel width ($2a$), imaginary parts of the normal mode frequencies decrease. However, for lower modes this effect is slightly more pronounced. This is because the time neces-
sary for the waves to travel over the open ocean across a distance of half channel width $a$ is equal to $a/\sqrt{D+1}$ and when this time is much less than the period of a longitudinal oscillation within the sloping channel, the geometrical spreading in the open sea is very efficient. This makes the free surface in the vicinity of the channel mouth almost flat due to the fast escape of the waves, rendering the channel mouth boundary condition effectively a Dirichlet condition with $\eta = 0$. Consequently, the waves reaching the mouth reflect very efficiently back towards the shore, limiting the radiation damping.

The rays that the waves follow in the sloping channel are straight lines at the shallower parts and they bend towards the corners as they get near to the mouth of the channel, due to geometrical spreading (see the streamlines in Figure C.1). If the width of the channel increases, this corner effect will penetrate deeper into the channel, making the rays longer. Longer rays will decrease the frequencies of the free oscillations. This can be observed in Table 2. Overall, because of this ray bending the frequencies in *MODEL-2* are lower than those in *MODEL-1*. Having said that, for a fixed value of channel width, if $D$
increases, the discrepancy between *MODEL-1* and *MODEL-2* decreases because for $D \to \infty$ the free surface in open ocean side of the domain will be flat for all free modes for both *MODEL-1* and *MODEL-2*, making the behaviour the same within the channel.

Now let us turn our attention to the transient response for *MODEL-2*. As before, we shall model the transient response to an incident wave of the form $\eta^I(t + x/\sqrt{D+1})$ and, as the first step, we shall look at the particular case in which $\eta^I$ is a
Dirac's delta function. For the sloping channel, such a response for the potential $\varphi$ corresponds to the integration of equation (45) with respect to $\omega$. This integration reduces down into a residue summation which we shall explain below. Since we are not





interested in the $y$-direction dependence of the runup, we are only interested in $A_0$. But since they are inter-related through the boundary conditions, we do end up having to calculate $A_1, A_2, .... A_n$ as well, even though we do not need them individually for our physical interpretations.

Remember that in *MODEL-1* we had an analytical expression for $A_0$, therefore also for $\varphi$ (see equations (18) and (14)). For *MODEL-2*, on the other hand, we do not have a closed-form relation for $A_0$ except when $\omega \to 0$. Obviously, when $\omega \to 0$, the free surface over the slope becomes flat and in this regime $A_0(\omega)$ is approximately equal to $2\tilde{\eta}^I(\omega)/i\omega$ for *both MODEL-1* and *MODEL-2*. For the rest of the frequencies, $\omega_k$, that make $A_0$ singular, we simply calculate circular integrals around each $\omega_k$ on the complex plane. Consequently, for $\varphi$ we have

$$
\varphi = \left( 2 \int_0^{\lambda-2(1-\sigma)} dt_0 \eta^I(t_0, x=1) \right) + \sum_{k=-\infty, k\neq 0}^{+\infty} \oint_{c_k} A_0(\omega) J_0(2\omega\sigma) d\omega
$$
$$
\times \int_0^{\lambda-2(1-\sigma)} dt_0 \, \eta^I(t_0, x=1) \exp\left(i\omega_k(\lambda - t_0)\right) \tag{53}
$$

where $c_k$ are integral contours with infinitesimal radii around each $\omega_k$. This approach is much faster than calculating the frequency integrals exclusively along the real axis. This is because, on the real axis the integrand becomes oscillatory and the accuracy can only be sustained using small integration steps (in *Model 1* we used several tens of thousand integration points to keep the integral stable). This is not feasible because for each integration point, an integral equation (equation (51)) needs to be solved to calculate the relevant $A_0$. On the other hand, for each $\omega_k$, the complex integral $\oint_{c_k} A_0(\omega) J_0(2\omega\sigma) d\omega$ can be satisfactorily calculated using just four points because the integrand for $\omega \to \omega_k$ becomes proportional to $1/(\omega - \omega_k)$ which is not oscillatory. A further practicality of this approach for an operational activity such as predicting a storm surge runup is that these contour integrals are *independent* of the structure of the incident wave and therefore can be pre-calculated for a particular bay or channel geometry and be injected in during the operational calculations.

In Figure C.1 the rays not only bend towards the mouth of the channel but also they coalesce. This coalescence reflects certain features that relate to the energy exchange to and from the channel. The characterization of these energy fluxes can be described using Poynting vectors. In the linear theory, for the shallow water, the non-dimensional Poynting vector, $\mathbf{P}$, can be defined as

$$
\mathbf{P} = h(x)\eta\mathbf{v} \tag{54}
$$

where $h(x)$ is the undisturbed water depth, $\mathbf{v}$ is the fluid velocity and $\eta$ is the non-dimensional pressure perturbation across the entire depth. In the steady case, the solution over the sloping channel is $\varphi = A_0(\omega)J_0(2\omega\sigma)$. This can be split into two waves travelling in opposite directions as $J_0(2\omega\sigma) = (H_0^{(1)}(2\omega\sigma) + H_0^{(2)}(2\omega\sigma))/2$. Here $H_0^{(1)}$ and $H_0^{(2)}$ correspond to the waves that travel towards the coast and from the coast respectively. In the steady case, since the time average of the energy flux is zero, to monitor the reflected wave, we *switch off* the ingoing wave. The time-averaged Poynting vector associated with the reflected wave within the channel, together with the hypergeometric function representing the standing wave is given in Figure (9). Note




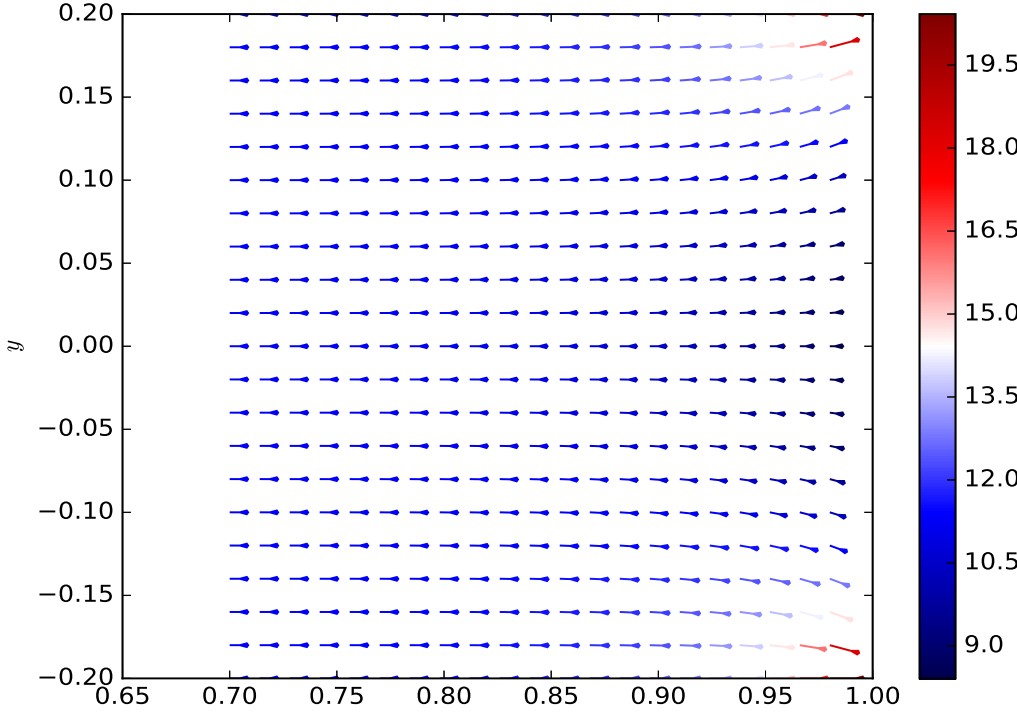

**Figure 9.** Time-averaged Poynting vectors (depth-integrated energy transfer), normalized to the square of the amplitude of incident wave, within the sloping channel with the incoming wave switched off. Channel half-width $a$ is 0.2, depth discontinuity $D$ is zero and the frequency of the incident wave is the real part of the first resonant frequency, $\omega_1$ (see Table 2).

that the time average of the hypergeometric function, coupled with the outgoing wave is not zero, because the Poynting vector is not a linear function of the wave.

For the open ocean, we calculated the time-averaged Poynting vector exclusively associated with the virtual source distribution at the mouth of the channel (Figure (10)). The reason this figure gives the impression that there is a net energy flux associated with the virtual sources is that the term $2\tilde{\eta}_0^I(\omega)\cos(\omega\frac{(x-1)}{\sqrt{D+1}})$ has been switched off. Finally, we include Figure (11) for the $x$-component of the Poynting vector along the mouth of the channel. This latter Poynting vector includes the full open ocean field. It integrates to zero because the energy density within the channel does not change with time when the steady regime is reached.





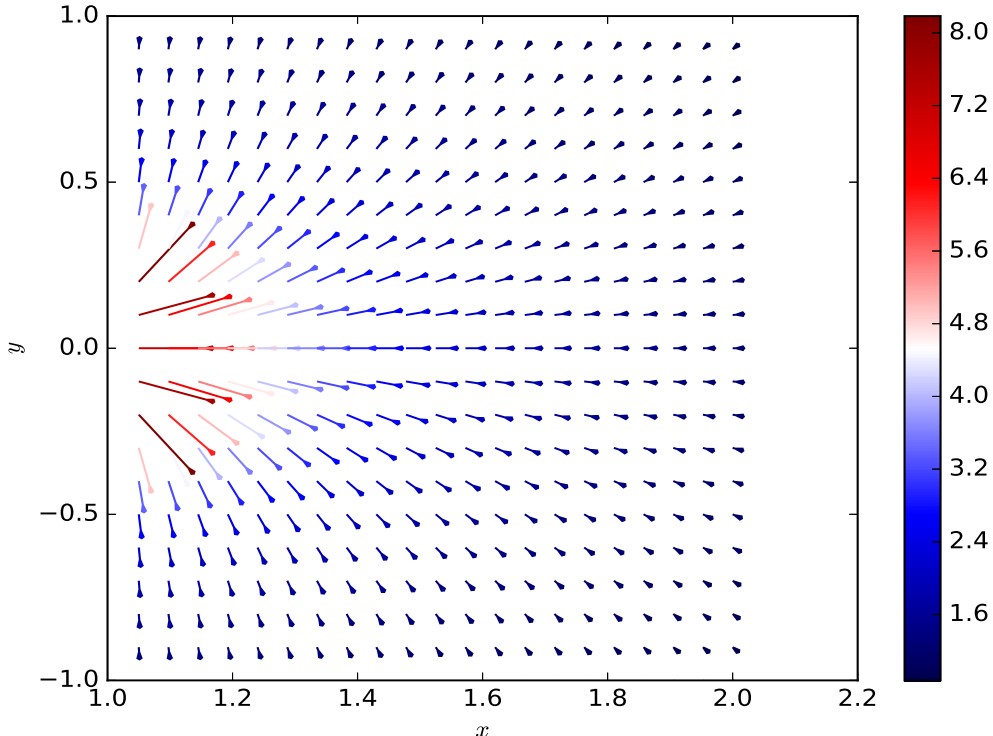

**Figure 10.** Time-averaged Poynting vectors exclusively associated with the virtual sources across the channel mouth. All parameters are the same as Figure 9.

## 6.1 The conformal mapping-based analyical solution of the integral equation for low-frequency range

We have already shown that $A_0(\omega) \approx 2\tilde{\eta}^I(\omega)/i\omega$ for wavelengths much larger than the length of the channel ($\omega \ll 1$). Using the conformal mapping the range of frequencies for which an analytical approximation of $A_0$ can be found, will be extended to $\omega \ll 1/a$ if depth discontinuity $D$ vanishes (see Mei et al. (2004) page 218). This second condition is much less restrictive

5    than ($\omega \ll 1$) for narrow channels.

Linearised shallow water equation for a monochromatic wave is Helmholtz equation

$$\left(\nabla^2 + \omega^2 / \text{depth}\right)\varphi = 0 \tag{55}$$

if the depth is constant. On a length scale much shorter than wavelength the solution of Helmholtz equation can be accurately approximated by the solution of Laplace equation ($\nabla^2\varphi = 0$, see Appendix C). The rays that the waves follow show a complex

10    pattern around the corner of the mouth of the channel. The problem will be greatly simplified if the solution of Helmholtz equation is replaced by that of Laplace equation in the zones affected by the corner effects. In the channel, the region with





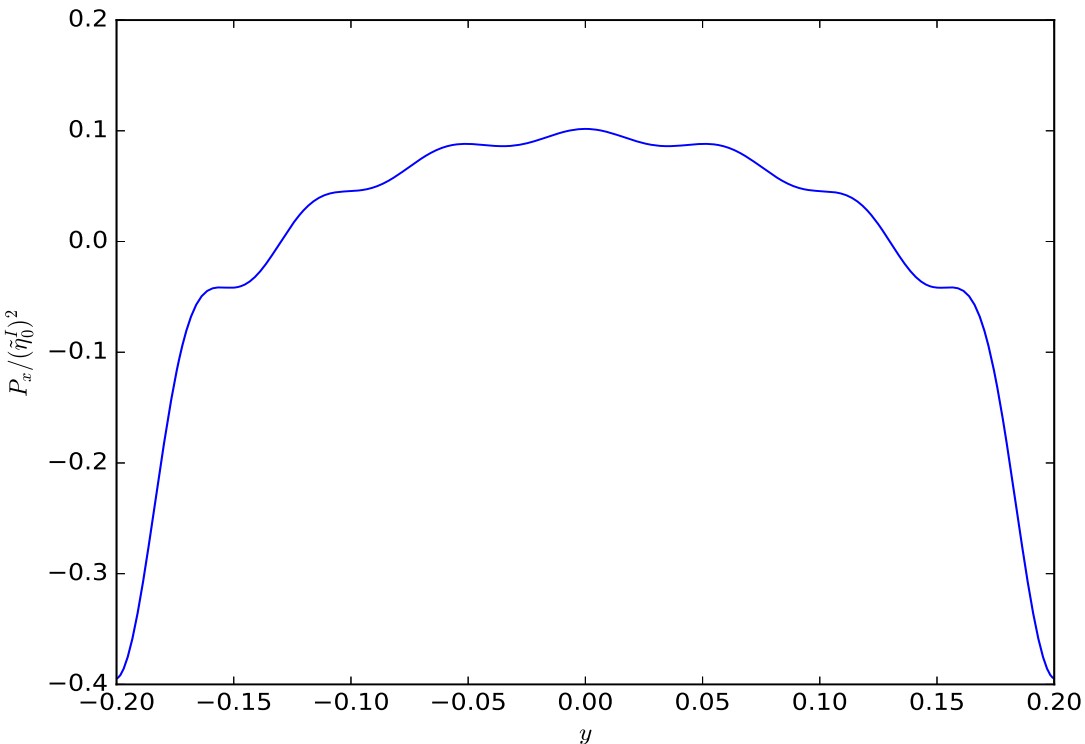

**Figure 11.** The $x$-component of time-averaged Poynting vectors across the channel mouth. This figure includes the full field given by equation (46). All parameters are the same as Figure 9.

$1 - x \gg a$ is not affected by the corner effects and the potential is then given by $A_0(\omega)J_0(2\omega\sigma)$. At intermediate ranges where the distance to the mouth is much smaller than the wavelength but larger than width $2a$, a linear approximation for the potential can be obtained

$$\varphi_\omega(x) \approx A_0(\omega)\left(J_0(2\omega) - \omega J_1(2\omega)(x-1)\right) \tag{56}$$

5 using the truncated Taylor series expansion of $A_0(\omega)J_0(2\omega\sqrt{x})$ around $x = 1$. Note that this linear potential is the solution of Laplace equation ($\nabla^2\varphi = 0$) in one dimension. Accordingly, the approximation given in equation (56) is only valid in the region where the relative change of depth is small.

In the open sea, when the distance from the mouth of the channel is much larger than the width of the channel the potential becomes, approximately for $D = 0$,

10 $$\varphi \approx 2a\frac{\tilde{S}_0(\omega)}{-2i}H_0^{(2)}\left(\omega\sqrt{(x-1)^2 + y^2}\right) + 2\tilde{\eta}_0^I(\omega)\frac{\cos\left(\omega(x-1)\right)}{i\omega} \tag{57}$$




since the distances between the target and the sources become approximately constant in the integral given by (46). Hence the Hankel function can be taken out of the integration and the only non-vanishing contribution to the integral will come from $\tilde{S}_0(\omega)P_0(\tilde{y}/a)$. Again, in the intermediate region where the distance to the mouth of the channel is much smaller than wavelength but larger than $2a$, equation (57) can be approximated by harmonic functions (solutions of Laplace equation)

$$5 \quad \varphi \approx 2a\frac{\tilde{S}_0(\omega)}{-i\pi}\left(-i\ln\left(\Gamma\frac{\omega}{2}\sqrt{(x-1)^2+y^2}\right)+1\right) + 2\frac{\tilde{\eta}_0^I(\omega)}{i\omega}. \quad (58)$$

where $\Gamma$ is the exponential of $\gamma = 0.5772...$ (Euler-Masheroni constant) and the term in the parenthesis is the first term in the series expansion of the Hankel function for the small argument.

As for the matching, a solution of Laplace equation will insure a smooth transition from the solution in the channel (see (56)) to the outer solution in equation (58). By the smooth transition it is meant that $\varphi$ and its derivative are continuous across the transition zone. In general a solution of Laplace equation is determined by one condition on the boundary. Here we have *two* conditions: one relating to the value of $\varphi$ and the second relating to its derivative. These two conditions make the system over-determined and it can be solved only if there is a special relation between $A_0(\omega)$ and $\tilde{\eta}^I(\omega)$ which reads as

$$A_0(\omega) = \frac{2\tilde{\eta}^I(\omega)}{i\omega}\left[J_0(2\omega) + a\omega J_1(2\omega)\left(\frac{-2}{\pi}\ln\left(\frac{e\pi}{2\Gamma\omega a}\right)+i\right)\right]^{-1}, \quad (59)$$

see Appendix C for its derivation. From (59) we can calculate the lower-mode resonant frequencies as those values of $\omega$ that cause $\omega A_0(\omega)$ to diverge. In Table 2, these values are compared with those calculated by minimizing the penalty integral given in (51), again for $D = 0$. One can see from (59) that there is strong resonance when $a$ is small (the so-called harbour paradox, Miles and Munk (1961)) because as $a \to 0$ the resonant frequencies approach the roots of Bessel function $J_0$ and consequently, when $J_0(2\omega)$ approaches zero $A_0(\omega)$ becomes proportional to $a^{-1}$. When the mouth width becomes too small, viscous effects start to play a role and the resonance would diminish, however we do not model this effect. Note that (59) can not be used to calculate the transient response of the system because the calculation of the transient involves a residue summation which invariably involves higher modes but (59) fails when the wavelength becomes of the same order of magnitude as the channel width, because when this happens Laplace equation can not be used as a substitute for Helmholtz equation. To see this failure at high modes, see Figure 13. Still, (59) is useful to calculate magnitude of runup as $t \to \infty$. In Figure (12) it is seen that the transient regime is very short-lived, in 7-8 oscillations the runup response converges approximately to the limiting amplitude. As for the nonlinearity, it is clearly seen that when the non-dimensional amplitude of the incoming wave is about 0.03 the relation between $\lambda$ and $t$ ceases to be one-to-one and the analysis is no longer valid.

According to equation (15), the maximum runup for the steady case is $|\omega A_0(\omega)|$. The figure (13) shows this value as a function of twice the frequency of the incident wave, calculated using both the integral equation and conformal mapping approaches. Kajiura (1977) also used conformal mapping to reach the same conclusion (note that his equivalent figure includes the amplification factor rather than the runup, therefore his values are half of those reported here).





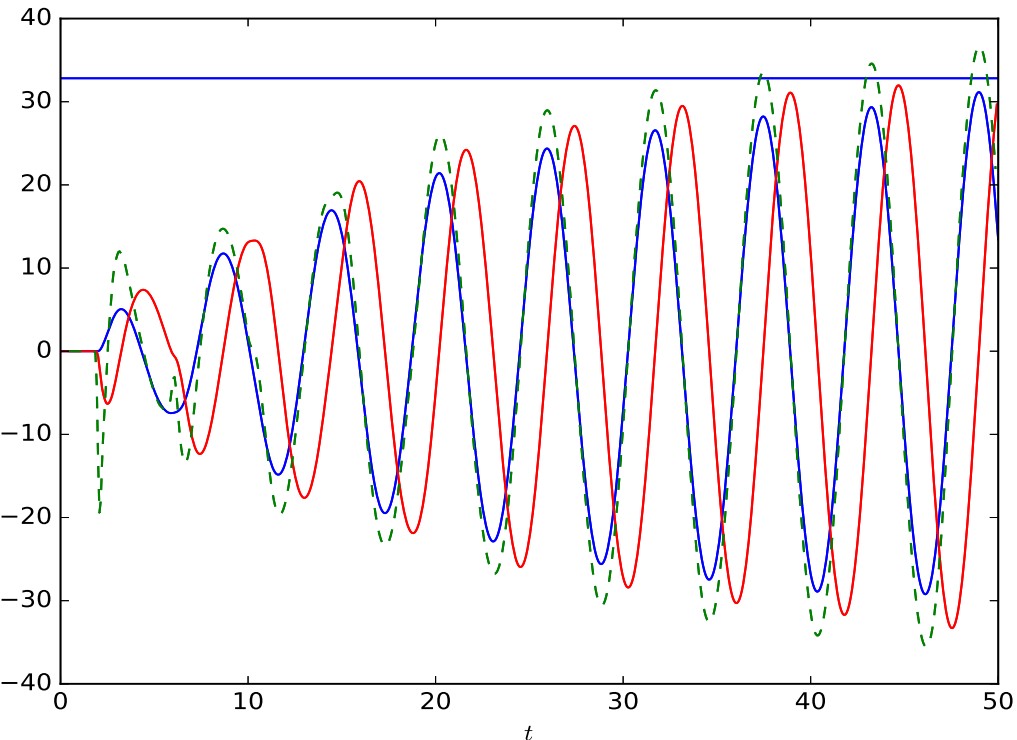

**Figure 12.** The blue continuous curve is the runup (calculated using equation (53)) normalized by the incident wave amplitude as a function of time. The channel width, $2a$ is equal to 0.2. The incident wave is given as $\eta_0^I \sin\left(\Re\omega_1(t+(x-1))\right)\theta(t+(x-1))$ where $\omega_1$ is the lowest free mode frequency for *MODEL-2* (see Table 2 for $D=0$), smoothed by multiplying it with the $tanh$ fuction as in figure (7). The continuous red curve is the shoreline velocity. The dashed green curve is the shoreline acceleration. The horizontal blue line is the maximum runup calculated using conformal mapping.

## 7 Conclusions

In this work we studied the resonance aspect of the coastal runup as a response to incident waves. The analysis follows a normal mode approach and examines the sensitivity of those normal modes to a given incident wave to produce coastal runup. In *MODEL-1*, significant runup sensitivity, in other words resonance, occurs only when $D$ is large. Large values of $D$ are not encountered very often in the nature, not even in the shelf breaks. In the two-dimensional open ocean and finite-width sloping channel case (*MODEL-2*) resonance occurs when the aspect ratio of the bay (width/length) is small. This kind of bay (or channel) geometry exists in many coastal regions, making the results relevant in engineering practice.

The residue method developed here can actually be generalized for more complicated channel geometries (such as piecewise constant slopes with varying width) by performing a "fusion" of this method with the boundary elements technique. This is





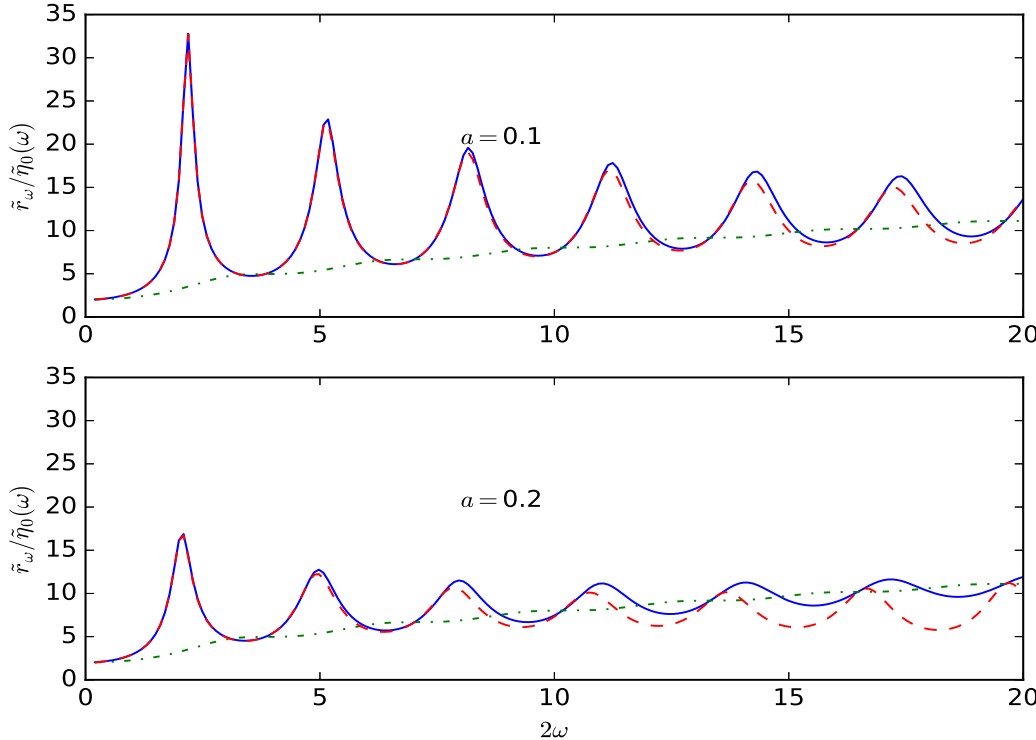

**Figure 13.** The maximum runup normalized to the amplitude of the incident wave for the steady case for two different channel half-width values. The continuous blue curves are computed using the integral equation and the red broken curves are obtained from the conformal mapping (see equation (59)). The green dot-dashed curve is the maximum runup for the infinitely wide channel. In the top figure the half-width of the sloping channel is 0.1, and in the bottom figure the half-width is 0.2.

because boundary elements technique is recently proving very efficient for solving Helmholtz equation in multiple dimensions (Gumerov and Duraiswami (2008), Takahashi and Hamada (2009)). So the fusion should work as follows: for a series of complex frequencies, the boundary elements determines those frequencies for which the response diverges (for a single run, the boundary elements, formulated in the temporal Fourier domain, can only calculate the response for a *monochromatic* incident wave. These are the normal mode frequencies for the particular geometry chosen. Then for any incident wave train (this can be, for instance, buoy data time series for an incoming Tsunami) we can calculate the contour integrals in (53) to compute the response much faster than inverse Fourier transforming the boundary element results. This computational economization has two advantages, the first is that these contour integrals can be pre-calculated and stored before, so during an operational emergency the only data that is needed is the wave train and the response can be calculated very fast (the last integral in (53)).


The other advantage is that the number of points necessary to numerically evaluate the contour integral is very low compared the inverse Fourier transformation along the real axis.

**Appendix A: Asymptotic approximation of the frequencies of the free modes for large $D$ in *Model-1***

In this appendix, the natural frequencies of *Model-1* will be evaluated for large $D$ using an asyptotic approximation. Note that the assumptions made in *Model-1* will hold as long as the lowest natural frequency of the system, $\omega_1$, is significantly larger than $a/\sqrt{D+1}$. The frequency of the fundamental mode is approximately 1.2 for large $D$. Accordingly, the condition $\omega_1 >> a/\sqrt{D+1}$ becomes, in terms of dimensional quantities, $1.2\sqrt{\alpha}/L' >> a'/\sqrt{D'+\alpha L'}$.

The matrix, $\mathbf{M}$, in equation (17) is

$$\mathbf{M}(\omega) = \begin{pmatrix} J_0(2\omega) & -1/i\omega \\ -\omega J_1(2\omega) & \sqrt{D+1} \end{pmatrix} \tag{A1}$$

The determinant of $\mathbf{M}$ is zero if $i\sqrt{D+1}J_0(2\omega) - J_1(2\omega)$ vanishes.

Expanding natural frequencies in powers of $\varepsilon = 1/\sqrt{D+1}$, we obtain

$$\omega_k = \omega_k^{(0)} + \Delta\omega_k = \omega_k^{(0)} + \varepsilon\omega_k^{(1)} + \varepsilon^2\omega_k^{(2)} + ... \tag{A2}$$

where $\omega_k^{(0)}$ is the unperturbed root with $J_0\big(2\omega_k^{(0)}\big) = 0$, and expanding it up to the term $\varepsilon^2$ and substituting into the determinant, one successively obtains, up to the second order,

$$\omega_k^{(1)} = i/2 \tag{A3}$$

$$\omega_k^{(2)} = \frac{1}{4}\frac{J_1{}'(2\omega_k^{(0)})}{J_0{}'(2\omega_k^{(0)})} \tag{A4}$$

where the primes mean the derivative with respect to the argument, $2\omega_k^{(0)}$. Consequently,

$$2\omega_k \approx 2\omega_k^{(0)} + \frac{i}{\sqrt{D+1}} + \frac{1}{2(D+1)}\frac{J_1'\big(2\omega_k^{(0)}\big)}{J_0'\big(2\omega_k^{(0)}\big)} \tag{A5}$$

is obtained for twice the preturbed normal mode frequencies.

**Appendix B: A simple, FFT-based approach for *MODEL-1***

In this section we introduce a simple approach to the incident wave runup using Fast Fourier Transformations (FFT) for *MODEL-1*. The solution procedure starts by a discrete sampling of the incident waveform at the toe of the slope on $N$ separate





times, $t_n = Tn/N$, for $n = 0, 1, ..., N-1$ with $T$ being the length of the sampling duration. In this particular example we use 262144 points to sample a sinusoidal waveform. Since the FFT procedure produces the replica of the finite-length signal one after another, the transient nature of the signal is lost to this artefact. In order to prevent this, the incident wave signal that we consider is set to zero for $t > t_c$ for which $T - t_c$ is large enough for the reflected waves to cease. Because of the application

of the FFT, the replicas keep to be produced but the consequent waves do not *see* the tails of the previous ones.

The Fourier transform of the runup is given by $i\omega A_0(\omega)$ where $A_0(\omega)$ is given in (18) which, at discrete frequencies $\omega_n = 2\pi n/T$, reads

$$A_0(\omega_n) = \frac{2\sqrt{D+1}}{2\pi\omega_n \left(i\sqrt{D+1}J_0(2\omega_n) - J_1(2\omega_n)\right)} \mathbf{FFT}(\eta^I(t)) \frac{T}{N} \tag{B1}$$

where $\mathbf{FFT}(\eta^I(t))$ is the Fast Fourier Transform of the incident waveform.

The FFT of the incident wave signal can be performed very fast using the Python routine scipy.fft. The first step is to produce a vector $\mathbf{w} = [\,\omega_1, \omega_2, ..., \omega_N\,]$ containing discrete frequencies. To eliminate the aliasing problem we need to consider the Nyquist frequency (see Press et al. (2007), page 605) which is half of the sampling rate of the signal. To do this we simply modify those elements of $\mathbf{w}$ in the following way:

$$\omega_n = \begin{cases} \frac{2\pi n}{T} & n = 0, 1, ..., N/2 \\ \frac{2\pi n}{T} - \frac{2\pi N}{T} & n = N/2 + 1, ..., N-1. \end{cases} \tag{B2}$$

The runup, $r$, at the discrete time step, $t_n$, is then equal to

$$r(t_n) = \frac{2\pi N}{T} \mathbf{IFFT}\left(i\omega_n A_0(\omega_n)\right) \tag{B3}$$

where $\mathbf{IFFT}$ stands for the Inverse Fast Fourier Transform.

## Appendix C: Conformal mapping for *MODEL-2*

The problem of of incident wave into a rectangular bay of uniform depth was solved in Mei et al. (2004), page 218. We

will extend his solution to the case of the sloping channel using the same conformal mapping. Actually, Kajiura (1977) also mentions a conformal mapping but its mathematical details are not included in his paper, neither elsewhere in publicly available literature.

Helmholtz equation is *not* invariant under the conformal mapping but on a length scale much shorter than the wavelength its solution can be approximated by that of Laplace equation. If the depth is equal to unity, then Helmholtz equation (55) will

become

$$\nabla^2\varphi + \omega^2\varphi = 0 \,. \tag{C1}$$

This equation, under a change of scale $x^* = x/d$, $y^* = x/d$, is transformed into

$$\nabla^{*2}\varphi + d^2\omega^2\varphi = 0 \tag{C2}$$




and in the limit $d \to 0^+$ it becomes Laplace equation ($\nabla^{*2}\varphi = 0$).

The complex function,

$$f^{-1}(\tilde{z}) = 1 + \frac{2a}{\pi}\left[-i\sqrt{\tilde{z}^2-1} + \ln\left(\frac{\tilde{z}}{\sqrt{\tilde{z}^2-1}+i}\right)\right] \tag{C3}$$

where $x + iy = f^{-1}(\tilde{z})$, maps the upper complex plane onto a semi-infinite half-plane connected to a semi-infinite channel

of width $2a$. The square root function used in this conformal mapping has a branch cut along the positive real axis with

$\sqrt{|\tilde{z}| \pm 0^+ i} = \pm\sqrt{|\tilde{z}|}$. Accordingly, $f^{-1}$ maps the segment $\zeta + i0^+$ for $1 > \zeta > 0$ on one side of the semi-infinite channel and

the segment with $0 > \zeta > -1$ is mapped on the other side.

In the $\tilde{z}$ space the solution of Laplace equation is

$$\varphi = m\ln(|\tilde{z}|) + c \tag{C4}$$

and it satisfies the solid boundary condition ($\partial_{\tilde{y}}\varphi = 0$ for $\tilde{y} = 0^+$ and $\tilde{x} \neq 0$). Note that $f^{-1}$ will map the origin to $-\infty$ (see

Figure C.1). The way a series of horizontal lines approaching the real axis are transformed by $f^{-1}$ is displayed in Figure C.1.

For a small argument, $f^{-1}(\tilde{z})$ is approximately equal to

$$z = x + iy \approx 1 + \frac{2a}{\pi}(1 + \ln(\tilde{z}/(2i))) \tag{C5}$$

with $x - 1 << -a$. Taking the exponential of both sides of C5, $\tilde{z}$ can be calculated as a function of $x$ and $y$. Therefore, the

solution of Laplace equation $m\ln(|\tilde{z}|) + c$ in term of $z = x + iy$ becomes

$$m\frac{\pi(x-1)}{2a} - m\ln(e/2) + c \, . \tag{C6}$$

Matching this with the solution of the wave equation $A_0(\omega)J_0(2\omega\sqrt{x})$ at $x = 1^-$ (note that $x = 1^-$ means that the distance to

the mouth is much smaller than the channel length but much larger than $a$, the width of the channel), we obtain

$$A_0(\omega)J_0(2\omega) = -m\ln(e/2) + c \tag{C7}$$

$$-A_0(\omega)\omega J_1(2\omega) = \frac{m\pi}{2a}. \tag{C8}$$

Regardless of the cut of the square root function, the magnitude of $\tilde{z}/(\sqrt{\tilde{z}^2-1}+i)$ tends to one for $|\tilde{z}| \to \infty$ (the cut of square

root affects only the argument of $\lim_{|\tilde{z}|\to+\infty} \tilde{z}/(\sqrt{\tilde{z}^2-1}+i)$). Accordingly, for large $|\tilde{z}|$, $z = f^{-1}(\tilde{z})$ becomes approximately

$-(2ai/\pi)\tilde{z}$. Matching $m\ln(\tilde{z}(z)) + c$ with (58)

$$\frac{2a\tilde{S}_0(\omega)}{\pi} = m \tag{C9}$$


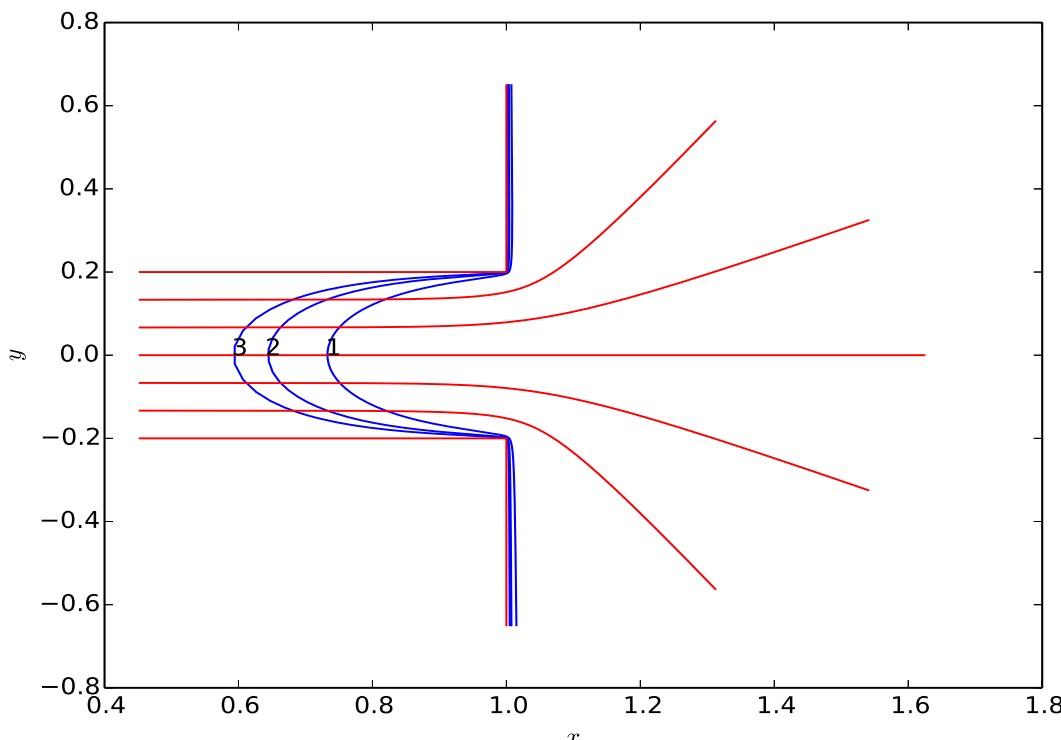

**Figure C.1.** The blue curves are $x + iy = z = f^{-1}(\tilde{z} = \zeta + \epsilon_n i)$ for $\zeta$ varying between -5 and 5 and $\epsilon_n = 0.06/n$. The three blue curves correspond to $n = 1, 2, 3$. The smaller the parameter $\epsilon_n$ becomes, further left these blue curves will reach (the left extremities of these curves are approximately $\frac{2a}{\pi} \ln(\epsilon_n)$ where $a$ is 0.2). The red curves are the streamlines.

$$a\tilde{S}_0 i + \frac{2a\tilde{S}_0}{\pi} \ln\left(\frac{\Gamma\omega}{2}\right) + \frac{2\tilde{\eta}^I(\omega)}{i\omega} = m\ln\left(\frac{\pi}{2a}\right) + c \tag{C10}$$

are obtained. Solving equations C7, C8, C9 and C10

$$A_0(\omega) = \frac{2\tilde{\eta}^I(\omega)}{i\omega}\left[J_0(2\omega) + a\omega J_1(2\omega)\left(\frac{-2}{\pi}\ln\left(\frac{e\pi}{2\Gamma\omega a}\right) + i\right)\right]^{-1} \tag{C11}$$

5   is found. The linear runup is then calculated by multiplying $A_0$ by $i\omega$.

Note that in $\tilde{z}$ space the equipotentials are upper semi-circles because the potential is proportional to $\ln(|\tilde{z}|)$ according to C4. Again, in the mapped space, the streamlines would be orthogonal to these equipotential ($\arg(\tilde{z}) = $ constant). Using C3, the streamlines are mapped onto the physical space and are shown as the red curves in Figure C.1.



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
