# Peer review of "On the Resonance Hypothesis of Storm Surge and Surf Beat Runup"

_Natural Hazards and Earth System Sciences, 2016_

## Referee Comment (RC1) · Anonymous Referee #1 · 21 Nov 2016

**Review of JFM 16-s-227-R1**
**On the resonance hypothesis of coastal runup**

Although motivated by tsunami which is the consequence of short-lived forcing, this work is an extension of existing work on resonance induced by persistent incident waves of a single period. On the latter topic extensive linear and nonlinear theories exist which appear unknown to the authors. By extensive citation of what has been done in the tsunami literature they overstate the similarity and understate the differences.

The paper deals with two idealized geometries : Model I is on the 1 D resonant scattering by a sloping shelf at the end of a long channel. This topic is a direct extension of shelf resonance whose physics by linear theory is well known, (see e.g., Longuet-Higgins for a circular shelf and Mei 1983 *Applied Dynamic of Ocean Surface Waves*). Model II is on 2 D scattering by a narrow channel open to the sea. When the channel depth is constant and equal to that of the outer sea, the linear and nonlinear resonance mechanics can be found in Bowers,JFM and Mei (1983). To treat coastal runup the authors choose the channel to be a beach of constant slope. The nonlinear solution of Carrier/Greenspan is then used. Outside the sloping shelf the sea depth is constant. The linearized theory is used. For both Models matching at the shelf break is done by linearizing the Carrier-Greenspan solution in an ad hoc manner without checking whether nonlinearity is locally important.

Throughout the paper the authors made extensive reference to past works on tsunami without considering the difference from their own work of persistent and periodic forcing . Tsunami is strictly transient where a finite number of leading crests are the most important to runup. In contrast resonance by periodic forcing takes a long time to reach quasi steady state which is important only at the end rather than the beginning. Hence the frequent and elaborate citations are mere digressions which only interupts their line of reasoning. Here are a few examples (1) The long paragraph on p 6 starting

with "Recently Stephanakis..." (2) The paragraph on p 8 " In an effort to calculate solitary wave runup...), and (3) The paragraph on p 9 starting with " The geometry considered by Stephanakis...". And many more. They can all be shortened or eliminated.

In my last review I suggested the authors to compare their nonlinear theory with a linear theory for physical implications in quantity and quality for both Models. Instead in the new section 4.2 the authors claim first that the importance of nonlinearity can be represented as a function of $u^2$. Without specifying the function they simply plotted $u^2$ itself. This is too indirect and very unsatisfactory.

For a physically straightforward topic the mathematical treatment here is very convoluted and can be more systematically presented. For example it would be clearer if the Green function in eq 3.9 is defined by listing the governing equation and boundary/initial conditions. For both Models I and II I failed to see any qualitatively new information different from a linear theory after the elaborate mathematics. Does a linearized theory not reveal the essential features in figures 4,5,6, 12 and 13? The authors never question whether in this nonlinear system chaos can be induced by simple harmonic incident waves. For Model II the discussion on the energy flux and Poynting vectors is not worthwhile since figs 9,10 and 11 are hardly exciting.

In short, in this revision the authors merely defended themselves without materially improving the original version. Instead of streamlining the already lengthy document, the paper has grown from 28 to 32 pages. The objectives and results are hardly related to their motivation. It would have been simpler to just make explicit comparison of runup by linear theories of resonant scattering by periodic incident waves and the final steady state (if any) from the nonlinear theory.

I regret that I am less inclined than last time to recommend the paper for consideration by JFM.

---

## Referee Comment (RC2) · Anonymous Referee #2 · 20 Dec 2016

<General comment>

The paper is devoted to coastal resonance of long waves which is particularly relevant to the problem of tsunamis. The authors investigate the natural frequencies for 1D and 2D idealized bathymetric settings that consist of a uniform slope connected to a flat bed, taking "radiation damping" into account. To highlight the role of the wave radiation, they introduce a discontinuity with varying depth into the jointing point of the bathymetry. The problem is mathematically reduced to determination of complex zeros of Bessel functions via residue theorem. They solve the problem through the asymptotic approach as well as Muller method and obtain the complex frequencies together with runup amplification factor. Comparing the results from different bathymetric settings, they discuss how the coastal resonance is excited by monochromatic incident waves. The fundamental study provides new insight into coastal resonance phenomena, clarifying the role of the wave radiation. However, I feel that the paper is lengthy and cumbersome. The key insight is difficult to single out. Also, it is hard to follow the mathematical derivation in some places due to lack of information. Therefore, I suggest the authors to revise the paper for more clarity and conciseness.

<Specific comments>

1. overall: The merit of using CG transform is not clear. The authors do not discuss much about the effect of nonlinearity in the sloping part of the bathymetry. I think the key results of this paper can be described more simply and concisely with linear models without using CG transform.

2. P.3 L23: The model bathymetry in Figure 1 is introduced without any explanation regarding the discontinuity. Please briefly explain the aim of introducing the discontinuity here. It would be easier for readers to understand the latter sections.

3. P.4 L1: The authors point out the model relevance to the storm surges in Tokyo bay. The storm surge in the semi-enclosed bay is generated by continuous forcing by wind stress. It is significantly developed when the typhoon track coincides with the bay axis. I think the case is not very relevant to the present model in which the wave is generated by short-time forcing out of the bay. Please take more relevant examples if the authors wish to keep "storm surges" in the title.

4. P.8 L25: Equation (22) may be (20). If so, please check the argument of the exponential function in (25).

5. P. 10 L11: I think this paragraph and Figure 2 can be omitted as it is distracting. The problem with two consecutive slopes is out of the initial model settings and seems not to be necessary for the latter discussion.

6. P. 15 L6: 18 => (18). Please follow the format of the journal.

7. P.15 L18: This us => This is

8. P.15 L13: The transient incident wave is initially given with Heaviside function, but it is later switched to tanh function on an ad-hoc manner to avoid discontinuities in wave profiles. I think this interrupts the flow of the discussion. As the authors mention, waves do not "switch on" at a given time in nature. The incident wave can be given with tanh function with a transitional scale from the beginning. The Heaviside case can be given with the zero transitional scale if it is really necessary for the discussion here.

9. Fig 5 and 6: I do not understand why the authors show the results of different modes in the two figures (1st mode in Fig 5 and 2nd mode in Fig 6). To see the effect of the discontinuity, the results of the same mode should be compared. Also, please clarify which of the two methods is used to obtain these results.

10. Fig 8: It is better to simply compare the linear and nonlinear wave profiles. It is not easy to know the quantitative difference from the present figure. However, I think this part can be omitted, if it just presents the nonlinear distortion of the time axis which is well-known to potential readers of this paper (This leads to my comment #1).

11. P.17 Section 5: I understand the role of this section, but the problem here is out of the initial model settings again. Please briefly describe the purpose of dealing with the infinite slope case at the beginning of the section.

12. P.17 L6: on => one.

13. P. 20 Eq (43): The authors need to describe the 2D governing equations, boundary conditions and approximations before presenting the analytical solution. The information is necessary to understand the following derivation.

14. Fig 9 and 10: Is it possible to combine these two figures? Then, we could see the overall picture of the energy flow over the 2D model bathymetry.

15. Section 7: The authors mention the future extension of the residue approach for engineering practices. However, the actual bathymetry and incident waves in nature are much more complicated. In practice, numerical models based on the 2D nonlinear

shallow water equations are widely used to predict long wave propagation and runup. The advantage of the proposed method is not clear in practical view point.

16. Title: I do not think the paper's title fits well with the contents. Please clarify what "hypothesis" the authors examine in the paper if they wish to keep the title.

---

## Referee Comment (RC3) · Anonymous Referee #3 · 22 Dec 2016

**Review of "On the resonance hypothesis of tsunami and storm surge runup"**

**by Postacioglu, Özeren and Canli**

The authors discuss the resonance hypothesis which has been proposed and investigated in a number of recent papers by e.g. Stefanakis, Dias and Dutykh (2011), Ezersky, Abcha & Pelinovsky (2013a-b), and Stefanakis, Xu, Dutykh and Dias (2015). In this connection, the authors emphasize the importance of radiation damping (partial reflection/transmission), which was ignored by the previous publications. I entirely agree with the authors, that this is an essential mechanism which needs to be addressed and incorporated. In the previous works (listed above), the reflection from the shoreline was modelled (physically or numerically) by locating the offshore boundary condition at a nodal point of the standing wave system. This actually makes little sense and has nothing to do with resonance. Hence the motivation for the present work is clear and relevant.

Unfortunately, the paper calls for some improvements. Generally, the conclusions tend to drown in mathematics and the paper is much too long. The more details you provide, the less clear is the overall conclusion. Several sections could easily be left out as for example section 4.2 (on nonlinear effects), which does not really add much to the overall picture. Also section 5 appears to be redundant mainly because a conclusion is never made. I am also not happy with section 6 dealing with the 2D ocean and a bay. This part would fit much better into a follow up paper.

I recommend that the authors make their conclusions and statements much more clear because I believe that the overall message deserves to come across to as many readers as possible. You do not achieve this but writing very lengthy papers filled with unnecessary mathematics.

Specific comments:

1) References to Stefanakis, Dias and Dutykh (2011); Stefanakis, Xu, Dutykh and Dias (2015) are not precise as some of the authors have been left out.
2) The authors use the notation "transient runup" in the sense that a monochromatic wave starts to be generated at a given instant. This is not the conventional definition of a transient phenomenon and the notation should be corrected accordingly.
3) Conclusions made on page 7, lines 5-10 are very much to the point. Bring them forward and make sure that they come across to the reader.

---

## Author Comment (AC1) · 19 Jan 2017

article

**Responses to the referee-I**

1. The referee refers to "Bowers,JFM and Mei (1983)" and says that when the channel depth is constant and equal to that of the outer sea, the linear and nonlinear resonance mechanics can be found in these texts.

   We disagree that Bowers (JFM) is relevant to our discussion because the resonance phenomenon in this article originates from a completely different mechanism in the sense that in Bower's work a larger semi-infinite channel connects to a finite-length narrow bay. In his case, the waves scattered towards offshore from the channel mouth bounce back from the sides of his finite-width wider channel. In our case the bay opens to a semi-infinite ocean and the scattered waves freely progress offshore. His problem is actually more complicated than ours, that is why he did some simplifying assumptions such as averaging over the width of the wider channel which is not relevant in our case. However, one of the referees in the JFM submission directed us to another work from seventies, Kaijura 1977 (K. Kajiura. Local behaviour of tsunamis. In D. Provis and R. Radok, editors, Waves on Water of Variable Depth, volume 64 of Lecture Notes in Physics, pages 72–79. Springer Berlin / Heidelberg, 1977.) which is very relevant to our case.

   We did use Mei's book for his conformal transform approach. However this technique is limited to long-wavelength (wavelength much larger than the channel width) and thus not applicable to find the transient response which involves *both*

high and low frequencies. So we resort to an integral equation approach and the compare the results with those obtained using conformal mapping (Figure 13 in the original submission).

2. The referee argues "The linearized theory is used. For both Models matching at the shelf break is done by linearizing the Carrier-Greenspan solution in an ad hoc manner without checking whether nonlinearity is locally important."

In the revised version we dedicate a new section to this issue (we also added an appendix). In both the original and revised manuscripts we have assumed a monochromatic small-amplitude incident wave. Near the resonant frequencies the amplitude of the standing wave over the slope is much larger than the amplitude of the incident wave. Therefore, in the revised version, using a perturbative approach, nonlinear effects are taken into account even in the deeper part of the channel. The boundary condition at the toe of the slope are accordingly perturbed. We limited ourselves to the first-order perturbation, so the frequencies are not affected. The important conclusion here is that the effect of nonlinearity arising from the boundary condition at the toe is 3 or 4 times smaller than the nonlinearity arising from shoaling. Thus, as far as the runup is concerned, the nonlinearity due to the boundary condition at the toe is not critical.

3. The referee mentions "Throughout the paper the authors made extensive reference to past works on tsunami without considering the difference from their own work of persistent and periodic forcing . Tsunami is strictly transient where a finite number of leading crests are the most important to runup. In contrast resonance by periodic forcing takes a long time to reach quasi steady state which is important only at the end rather than the beginning. Hence the frequent and elaborate citations are mere digressions which only interupts their line of reasoning. ". He/she adds "Here are a few examples (1) The long paragraph on p 6 starting with "Recently Stephanakis..." (2) The paragraph on p 8 " In an effort to calculate solitary wave runup...), and (3) The paragraph on p 9 starting with "

[Figure]

The geometry considered by Stephanakis...". And many more. They can all be shortened or eliminated."

The referee seems to be particularly insistent on persuading us not to cite Stephanakis' papers. We are not inclined to do so to keep the integrity of the manuscript which dwells very much on the discussions on the few recent works on runup resonance. Despite the addition of a new section and appendix, we did substantially shorten the manuscript (it is now 29 pages instead of 36 pages).

It is indeed true that a Tsunami is a transient but we do calculate the time to reach the steady-state and therefore can pinpoint strict time limits for the transient duration. We believe this is important (see Figure 12 in the original manuscript).

4. The referee mentions "or a physically straightforward topic the mathematical treatment here is very convoluted and can be more systematically presented. "

   We agree with this criticism and made an effort to shorten the mathematical derivations while, at the same time, keeping the manuscript's main important ideas.

5. The referee mentions "The authors never question whether in this nonlinear system chaos can be induced by simple harmonic incident waves."

   Chaos is a whole different topic which is not within the scope of the present manuscript. Furthermore, to chaos to ensue (if any), long durations are necessary which we do not consider.

---

## Author Comment (AC2) · 19 Jan 2017

article

**Responses to the referee-II**

The referee mainly points out that the manuscript is lengthy and that the key insight is difficult to single out. As a result of this and the criticisms brought in by the two other referees, we shortened the manuscript as much as possible. The referee also reports specific points for which our answers follow below.

1. The referee mentions " overall: The merit of using CG transform is not clear. The authors do not discuss much about the effect of nonlinearity in the sloping part of the bathymetry. I think the key results of this paper can be described more simply and concisely with linear models without using CG transform."

   We agree with the referee in the sense that the resonant frequencies that we calculate are independent from nonlinear effects, because we linearize the boundary conditions at the toe of the slope. We appended a sentence to clarify this out. However we want to keep the CG approach in order to be able to calculate the runup by taking nonlinear effects of shoaling into account. This is important from the hazard point of view.

2. The referee mentions "the model bathymetry in Figure 1 is introduced without any explanation regarding the discontinuity. Please briefly explain the aim of introducing the discontinuity here. It would be easier for readers to understand the latter sections."

We used the dicontinuity both to mimic a natural bathymetric setting and also to see the influence of the size of the discontinuity on resonance. As found, there is very little resonance for MODEL-I when the discontinuity is zero. We added a sentence to point this out.

3. The referee mentions "he authors point out the model relevance to the storm surges in Tokyo bay. The storm surge in the semi-enclosed bay is generated by continuous forcing by wind stress. It is significantly developed when the typhoon track coincides with the bay axis. I think the case is not very relevant to the present model in which the wave is generated by short-time forcing out of the bay. Please take more relevant examples if the authors wish to keep "storm surges" in the title."

In most cases yes but not all cases such as, for instance the Hurricane Katrina which followed a curved track, resulting in time-varying parameters. Perhaps "meteotsunamis" rather than "typical" storm surges fit better to our case. On the other hand. The wind forcing can generate Kelvin waves which are time-periodic. Such Kelvin waves can trigger the type of waves we are considering in the bays. This may or may not be termed as storm surge due to the non-specificity of the jargon. So if the editorial finds it fit, we can change the title from storm surges to meteo-tsunamis to be in a more relevant wave frequency band:

http://www.nat-hazards-earth-syst-sci.net/6/1035/2006/nhess-6-1035-2006.pdf

4. The referee mentions "Equation (22) may be (20). If so, please check the argument of the expo- nential function in (25)"

There is indeed an inconsistency. The equations (20) and (22) are correct (equation numbering according to the original manuscript). But the $t_0$ term in the exponential term in equation (23) must be cancelled as it is already an argument of the Green's function. Actually the same inconsistency persist in the equation

(24) but the result in equation (25) is correct. We fixed this inconsistency in the revised manuscript.

5. The referee mentions "I think this paragraph and Figure 2 can be omitted as it is distracting. The problem with two consecutive slopes is out of the initial model settings and seems not to be necessary for the latter discussion."

   We eliminated the two consecutive slopes case completely in the revised manuscript.

6. "P. 15 L6: 18 $=>$ (18). Please follow the format of the journal."

   corrected

7. "This us $=>$ This is"

   corrected

8. The referee mentions ".15 L13: The transient incident wave is initially given with Heaviside function, but it is later switched to tanh function on an ad-hoc manner to avoid discontinuities in wave profiles. I think this interrupts the flow of the discussion. As the authors mention, waves do not "switch on" at a given time in nature. The incident wave can be given with tanh function with a transitional scale from the beginning. The Heaviside case can be given with the zero transitional scale if it is really necessary for the discussion here."

   In previous studies, such as Stefanakis et al.(2015) only $\eta$ and runup curves have been displayed. Shoreline velocities have been displayed using colour rather than curves. This causes the velocity discontinuity not to show up visually in the graphs (see Fig(6) of the mentioned paper). From the figure it is obvious that the derivative of the runup with respect to time is discontinuous. So we did not smooth the incoming wave -in the linear model-when runup is displayed as a function of time so that future researchers can compare their results with us

without having to introduce a smoothing parameter. On the other hand, since the fluid velocity ,$u$, is intrinsically discontinuous when the wave starts abruptly at $t_0$ smoothing becomes unavoidable. Smoothing is an absolute necessity also for the nonlinear case because otherwise the CG transformation becomes discontinuous $(\lambda = t - u)$.

9. The referee mentions: " Fig 5 and 6: I do not understand why the authors show the results of different modes in the two figures (1st mode in Fig 5 and 2nd mode in Fig 6). To see the effect of the discontinuity, the results of the same mode should be compared. Also, please clarify which of the two methods is used to obtain these results."

   We agree. We now only give the results for the first mode. We also include the two figures in a single composite figure. We also modified the caption in order to eliminate the confusion by including the value of $D$ in parenthesis (the modes are obviously $D$-dependent). The method used is the residues, also mentioned in the new caption.

10. The referee mentions: "Fig 8: It is better to simply compare the linear and non-linear wave profiles. It is not easy to know the quantitative difference from the present figure. However, I think this part can be omitted, if it just presents the nonlinear distortion of the time axis which is well-known to potential readers of this paper (This leads to my comment 1)"

    Corrected as recommended by the referee

11. The referee mentions:"P.17 Section 5: I understand the role of this section, but the problem here is out of the initial model settings again. Please briefly describe the purpose of dealing with the infinite slope case at the beginning of the section."

    We added a justification paragraph at the beginning of the section. It basically points out the fact that most publications on runup deal with infinite slopes. We

included the section for comparison purposes for future research. Infinite slope runup for steady-state regime has been calculated by Pelinovsky and Mazova (1992), we generalized this to the transient regime. We also show that there is very little resonance even when the wavemaker sits on the node of the standing waves.

12. "P.17 L6: on => one."

corrected

13. The referee mentions: "P. 20 Eq (43): The authors need to describe the 2D governing equations, boundary conditions and approximations before presenting the analytical solution. The informa- tion is necessary to understand the following derivation."

We added the governing equations

14. The referee mentions: " Fig 9 and 10: Is it possible to combine these two figures? Then, we could see the overall picture of the energy flow over the 2D model bathymetry."

We had to remove all energy flux discussion due to the criticism of other referees. However the ray path figure clearly shows the concentration of the rays near the corners of the mouth where the energy fluxes are large.

15. The referee mentions "Section 7: The authors mention the future extension of the residue approach for engineering practices. However, the actual bathymetry and incident waves in nature are much more complicated. In practice, numerical models based on the 2D nonlinear shallow water equations are widely used to predict long wave propagation and runup. The advantage of the proposed method is not clear in practical view point."

We partially agree in the sense that a certain idealization is necessary (multiple constant slopes or a bay of parabolic cross-section). However the residues that

we calculate are independent of the character of the incident wave, so they can be computed beforehand and can be used later for any kind of wave for a given locality synchronously as the wave approaches.

16. The referee mentions: "Title: I do not think the paper's title fits well with the contents. Please clarify what "hypothesis" the authors examine in the paper if they wish to keep the title."

    We think that we are testing the hypothesis of the relevance of runup resonance. However we are of course open to any alternative suggestion for the title.

---

## Author Comment (AC3) · 19 Jan 2017

article

**Responses to the referee-III**

The general view of the referee is that the manuscript is too long and the main ideas are obscured by mathematical details. We significantly shortened the manuscript and shaved off some of the maths. The referee also rightly points out that the section-5 did not have a conclusion which we now added. We did not remove the nonlinear section due to the insistence of the first referee. However the resonant frequencies that we give in the tables are independent of nonlinearities. Another point raised was that the 2-D case would be better left to another paper. Well, we are more inclined to keep it because in realistic bathymetries the cases where the resonance is important are often bays (as is clear from the manuscript, in 1-D case, resonance becomes significant only for unrealistically high bathymetric discontinuities). However, following the advice of the referee, we considerably shortened the 2-D section. Below are the responses to the particular comments.

1. The referee mentions: "References to Stefanakis, Dias and Dutykh (2011); Stefanakis, Xu, Dutykh and Dias (2015) are not precise as some of the authors have been left out."

   Corrected

2. The referee mentions:"The authors use the notation "transient runup" in the sense that a monochromatic wave starts to be generated at a given instant. This is not

the conventional definition of a transient phenomenon and the notation should be corrected accordingly."

We removed the word "steady" as it can indeed lead to confusion. We mean standing waves of constant amplitude. Corrected. We used "transient" for the waves that occur before the settling of the standing wave regime. We explain this in the revised manuscript.

3. The referee mentions: "Conclusions made on page 7, lines 5 ‐ 10 are very much to the point. Bring them forward and make sure that they come across to the reader"

The referee-1 wants us to remove these lines but we want to keep them because, as suggested above by the referee comment, the gravity of the manuscript sits there.

---

## Author Response (AR1)

Text-referenced page and line-specific corrections for each referee (**the numbers correspond to the specific requests of the referees for a text change "as listed in our responses"**). Only the text-changes are reported here (for the general comments see the responses for each referee)

Overall, we made an effort to both shorten the manuscript and render the mathematics more easily tractable. We also made the main important results more visible. The original manuscript was 36 pages long, it is now 31 pages long despite the addition of a new section and a new appendix about the nonlinearity.

Referee-1

1. A detailed answer has been provided in the answers to the referee. As explained there, we refer now Kaijura 1977 (K. Kajiura. Local behaviour of tsunamis. In D. Provis and R. Radok, editors, Waves on Water of Variable Depth, volume 64 of Lecture Notes in Physics, pages 72–79. Springer Berlin / Heidelberg, 1977.) which is very relevant to our case (page-22, line 26 in the revised submission).

2. We added a whole new section on a detailed analysis of nonlinearity both due to shoaling and the boundary condition at the toe of the slope (section 4.2 and Appendix-B).

4. We shortened the manuscript, especially the mathematical formulation. We have shortened the section-3 (Green's function and free-mode expansion), the original submission went from page-5 to 12, the revised version goes from page 5 to 10 and the presentation is, we believe, clearer.

Referee-2

1. We clarify in the first paragraph in page-7.
2. An explanation for as to why we have the bathymetric discontinuity is added to the end of the abstract. The reason we added this to the abstract rather than the introduction is because the fact that the depth discontinuity is the main controlling factor for the resonance only becomes clear at the end of the analysis.
3. In page-3 line 29 we explain how storm surges can indirectly excite such modes through Kelvin waves.
4. corrected
5. We eliminated the two consecutive slopes case completely in the revised manuscript.
6. corrected
7.corrected
9. The old Figure-4 and Figure-5 have been merged to produce the new Figure-4 and on the request of the second referee in both cases (D= 0 and 5)  the same

mode has been plotted.

10. The new Figure-6 has been produced upon the request of referee-2. It is a modified version of the old Figure 8 (we eliminated the $\lambda$ -t plot).

11. New explanation has been added in the second line of the section-5 starting with "The reason for this practice..." (page-16, line 19)

12.corrected

13.Page-18 we added the governing equations and the boundary conditions (equation 38)

14.We had to remove all energy flux discussion due to the criticism of other referees. However the ray path figure clearly shows the concentration of the rays near the corners of the mouth where the energy fluxes are large.

Referee-3

1. corrected

2. We removed the word "steady" as it can indeed lead to confusion. We mean standing waves of constant amplitude. Corrected. We used "transient" for the waves that occur before the settling of the standing wave regime. We explain this in the revised manuscript (page-3, line 10).

3.The referee-1 wants us to remove these lines but we want to keep them because, as suggested above by the referee comment, the gravity of the manuscript sits there. Actually in the revised version (page-6, line 23-26 we give a slightly more detailed discussion).

Referee-3 was also of the opinion that the 2-D case should not be included in this paper but should be published as a follow-up paper. We prefer to keep it as it is a novelty and unlike the MODEL-1, it does not need a large D value to produce significant resonance. We also shortened the 2-D case significantly (from 9 pages in the original submission to about 6 pages in the revised manuscript).

[revised manuscript text omitted]

---

## Referee Report (RR1)

Comments on the manuscript entitled
**"On the Resonance Hypothesis of Tsunami and Storm Surge Runup"**
**(Ref. NHESS-2016-334-version3)**
by **Nazmi Postacioglu, M. Sinan Özeren & Umut Canlı**

This is a nice, semi-analytical paper that provides a useful analysis of the resonance of long wave runup on beaches. The paper is of obvious interest for the readers of *NHESS* and it is fairly well written. There are, however, some weaknesses that require a moderate revision, also along the lines below proposed, before publication be granted.

**Main issues**

**The seaward boundary conditions** Although linearization of the seaward boundary condition is an understandable simplification of the problem, this leads to some underestimation of the wave runup, as demonstrated by Antuono & Brocchini (2007). I am not requesting that nonlinear boundary conditions be used, which would make the analysis less straightforward, rather that the above be explicitly acknowledged.

**The model limitations** Beyond linearization of the seaward boundary condition, the model is, by force, characterized by a number of shortcoming, which, though understandable, should be properly acknowledged (see "Specific Issues").

**The context** The Introduction and various descriptions, though clear and fairly well structured, seem to be missing quite a few important contributions. These should be reintroduced and properly described as suggested, for example, in the "Specific Issues".

**Specific issues**

The following specific issues require attention (line numbers are those provided by the authors' editing):

1. Abstract and Conclusions. The model, though interesting, is characterized by some shortcomings (see in the following), which should be briefly acknowledged here;

2. Introduction, first 3 lines. This list of references could well accommodate the following works, all based on Carrier & Greenspan transformation but giving account of different issues, like the horizontally-2D nature of the flow, the wave groupiness and the actual boundary value nature (not initial value one) of the problem at hand: Brocchini (1998), Brocchini & Gentile (2001) and Antuono & Brocchini (2008);

3. page 3, lines 19-21. Fundamentals for this important result are discussed in Antuono & Brocchini (2007), which should be recalled here;

4. page 4, line 11. $\eta'$ is a free surface elevation, not a wave height (crest minus through elevation...);

5. page 6, lines 3-5. This is a fairly important matter because the mentioned linearization generally leads to a shoreline dynamics weaker than that it would be obtained from a more accurate data assignment, as demonstrated by Antuono & Brocchini (2007). This should be briefly but clearly acknowledged (here, in the Abstracts and Conclusions);

6. page 8, lines 1-2. The divergence here discussed, though not significantly affecting the solution in the shallow waters, is a mark of some shortcomings in the analysis of the problem. For the sake of clarity, this should be briefly acknowledged in both Abstract and Conclusions;

7. page 9, lines 6-8. This sentence calls for brief discussion of the applicability of the proposed theory to real-life conditions;

8. page 13, lines 9-10. This should be commented in conjunction to the previous point;

9. page 15, lines 1-3. This is another shortcoming of the model that, for the sake of clarity, should be briefly acknowledged in both Abstract and Conclusions;

10. section 4.2. This section, dedicated to the role of nonlinearities, should include a discussion of the improper use of linearized boundary conditions, leading to an incorrect weakening of the shoreline dynamics, as demonstrated by Antuono & Brocchini (2007);

11. page 15, line 6. This sentence, depends on the previous one and cannot stand alone;

12. page 18, line 17. Among the few analytical studies of this problem that actually provide useful insight in the physics at hand are those of Brocchini & Peregrine (1996) and Brocchini (1998), which should be recalled here.

**References**

- Antuono, M. & Brocchini, M.. (2007). The Boundary Value Problem for the Non-linear Shallow Water Equations. *Stud. Appl. Maths.* **119(1)**, 73-93;

- Antuono, M. & Brocchini, M. (2008). Maximum run-up, breaking conditions and dynamical forces in the swash zone: a Boundary Value approach. *Coast. Engng.* **55(9)**, 732-740;

- Brocchini, M. (1998). The run-up of weakly-two-dimensional solitary waves *Nonlin. Proc. Geophys.* **5**, 27-38;

- Brocchini, M. & Peregrine, D.H. (1996). Integral flow properties of the swash zone and averaging. *J. Fluid Mech.* **317**, 241-273;

- Brocchini, M. & Gentile, R. (2001). Modelling the run-up of significant wave groups. *Cont. Shelf Res.* **21(15)**, 1533-1550.

---

## Author Response (AR2)

Dear Professor Didenkulova,

Thank you very much for your comments and suggestions. We took into account all of them to modify our manuscript (including the comments by the fourth referee). In your comments you mention that Bowers' 1977 (JFM) paper is relevant to our article and that it should " in the limiting case of infinitely wide channel, give the result similar to ours". Bowers obtained an analytical result in closed form for a geometry where a flat-bottomed rectangular bay is connected to a channel of greater but finite width. Furthermore, he considered only the particular case (page 75, line 3 in his article) where the wavelength of the incident wave is larger than the width of the channel (in our case the "wider channel" has infinite width, the open ocean). Hence, to apply his theory to our case the wavelength of the incident wave has to tend to infinity. Both in our Model-1 and Model-2, the runup divided by the amplitude of the incident wave tends to two (2) when the incident wave frequency goes to zero. In Bower's paper, the relevant equation in page 75 gives the potential inside the bay and from this we see that his coefficient "D" is equal to half the runup. He also provides an explicit relation for "D" in the same page (before the equation (8)), in terms of the amplitude of the incident wave. In the long wave limit, D is equal to one, as you pointed out. If we apply his formula blindly for any finite frequency with the channel width being infinite, we get infinite runup at the resonant frequency which is obviously a misprediction because at a finite frequency there should be more than one radiating modes in the wider channel which his formulation can not model. In Bowers' model the wavenumber in y-direction takes discrete values due to the finiteness of the width of the channel, in our case the wavenumber in y-direction is continuous. In the re-revised version of our manuscript (in page-22, highlighted) we explain that we recover Bowers' result for the zero frequency.

We removed the word "tsunami" from the title as you saw fit. We do still mention Tsunamis within the text but only when refering other peoples' works. We also removed the word Tsunami from the title and limited the applications to storm surges and surf beats, both being long waves.

We provide a a separate letter to list the corrections that we did to address to the comments of Referee-4.

Sincerely yours

M Sinan Özeren

**Responses to Referee-4**

Below are the list of corrections that we did as response to the comments of the Referee-4:

1-Abstract and Conclusions. The model, though interesting, is characterized by some shortcomings (see in the following), which should be briefly acknowledged here;

2-Introduction, first 3 lines. This list of references could well accommodate the following works, all based on Carrier & Greenspan transformation but giving account of different issues, like the horizontally-2D nature of the flow, the wave groupiness and the actual boundary value nature (not initial value one) of the problem at hand: Brocchini (1998), Brocchini & Gentile (2001) and Antuono & Brocchini (2008);

*We include all the references pointed by the referee in the re-revised manuscript (highlighted in the PDF text)*

3- page 3, lines 19-21. Fundamentals for this important result are discussed in Antuono & Brocchini (2007), which should be recalled here;

*done, see lines 19-20, page 3*

4- page 4, line 11. $\eta'$ is a free surface elevation, not a wave height (crest minus through elevation...);

*corrected all occurrences throughout the text ( highlighted in the PDF text)*

5-page 6, lines 3-5. This is a fairly important matter because the mentioned linearization generally leads to a shoreline dynamics weaker than that it would be obtained from a more accurate data assignment, as demonstrated by Antuono & Brocchini (2007). This should be briefly but clearly acknowledged (here, in the Abstracts and Conclusions);

*We added new text to the nonlinear section to point out this issue with a new reference to Antuono & Brocchini (2007) (see lines 20-23, page 16). We also mention the effect of the linearity (highlighted) of the seaward boundary condition linearization in the abstract and conclusion where we give a reference to Antuono & Brocchini (2007).*

6- page 8, lines 1-2. The divergence here discussed, though not significantly affecting the solution in the shallow waters, is a mark of some shortcomings in the analysis of the problem. For the sake of clarity, this should be briefly acknowledged in both Abstract and Conclusions;

*done (highlighted)*

7- page 9, lines 6-8. This sentence calls for brief discussion of the applicability of the proposed theory to real-life conditions;

*Lines 19-20, page 9, we added a sentence expressing that although the effect is minor*

*in Model-1, it is important for Model-2, therefore the formulation remains relevant.*

8-page 9, lines 6-8. This sentence calls for brief discussion of the applicability of the proposed theory to real-life conditions;

*removed, see the previous comment*

9-page 15, lines 1-3. This is another shortcoming of the model that, for the sake of clarity, should be briefly acknowledged in both Abstract and Conclusions;

*In both the abstract and the conclusion we now explicitly mention that the incident wave is linear*

10-section 4.2. This section, dedicated to the role of nonlinearities, should include a discussion of the improper use of linearized boundary conditions, leading to an incorrect weakening of the shoreline dynamics, as demonstrated by Antuono & Brocchini (2007);

*Now mentioned in the nonlinear section*

11-page 15, line 6. This sentence, depends on the previous one and cannot stand alone;

*page 15, line 6 (these are line numbers in the previous version) reads:"As long as the waves do not break, the nonlinearity arising from the shoaling over the slope is accounted for by the CG".*

*We do not understand what is meant in this comment.*

12- page 18, line 17. Among the few analytical studies of this problem that actually provide useful insight in the physics at hand are those of Brocchini & Peregrine (1996) and Brocchini (1998), which should be recalled here.

*Both now referenced (highlighted)*